# The neuronal calcium sensor NCS-1 regulates the phosphorylation state and activity of the Gα chaperone and GEF Ric-8A

Daniel Muñoz-Reyes[1], Levi J McClelland[2], Sandra Arroyo-Urea[3], Sonia Sánchez-Yepes[4], Juan Sabín[5,6], Sara Pérez-Suárez[1], Margarita Menendez[7,8], Alicia Mansilla[4,9], Javier García-Nafría[3], Stephen Sprang[2], Maria Jose Sanchez-Barrena[1]*

[1]Department of Crystallography and Structural Biology, Institute of Physical-Chemistry 'Blas Cabrera', CSIC, Madrid, Spain; [2]Center for Biomolecular Structure and Dynamics, and Division of Biological Sciences, University of Montana, Missoula, United States; [3]Institute for Biocomputation and Physics of Complex Systems (BIFI) and Laboratorio de Microscopías Avanzadas (LMA), University of Zaragoza, Zaragoza, Spain; [4]Department of Neurobiology, Instituto Ramón y Cajal de Investigación Sanitaria, Hospital Universitario Ramón y Cajal, Madrid, Spain; [5]AFFINImeter Scientific & Development team, Software 4 Science Developments, Santiago de Compostela, Spain; [6]Departamento de Física Aplicada, Universidad de Santiago de Compostela, Santiago de Compostela, Spain; [7]Department of Biological Physical-Chemisty, Institute of Physical-Chemistry 'Blas Cabrera', CSIC, Madrid, Spain; [8]Ciber of Respiratory Diseases, ISCIII, Madrid, Spain; [9]Department of Systems Biology, Universidad de Alcala, Madrid, Spain

**\*For correspondence:** xmjose@iqf.csic.es

**Competing interest:** The authors declare that no competing interests exist.

**Abstract** The neuronal calcium sensor 1 (NCS-1), an EF-hand $Ca^{2+}$ binding protein, and Ric-8A coregulate synapse number and probability of neurotransmitter release. Recently, the structures of Ric-8A bound to Gα have revealed how Ric-8A phosphorylation promotes Gα recognition and activity as a chaperone and guanine nucleotide exchange factor. However, the molecular mechanism by which NCS-1 regulates Ric-8A activity and its interaction with Gα subunits is not well understood. Given the interest in the NCS-1/Ric-8A complex as a therapeutic target in nervous system disorders, it is necessary to shed light on this molecular mechanism of action at atomic level. We have reconstituted NCS-1/Ric-8A complexes to conduct a multimodal approach and determine the sequence of $Ca^{2+}$ signals and phosphorylation events that promote the interaction of Ric-8A with Gα. Our data show that the binding of NCS-1 and Gα to Ric-8A are mutually exclusive. Importantly, NCS-1 induces a structural rearrangement in Ric-8A that traps the protein in a conformational state that is inaccessible to casein kinase II-mediated phosphorylation, demonstrating one aspect of its negative regulation of Ric-8A-mediated G-protein signaling. Functional experiments indicate a loss of Ric-8A guanine nucleotide exchange factor (GEF) activity toward Gα when complexed with NCS-1, and restoration of nucleotide exchange activity upon increasing $Ca^{2+}$ concentration. Finally, the high-resolution crystallographic data reported here define the NCS-1/Ric-8A interface and will allow the development of therapeutic synapse function regulators with improved activity and selectivity.

## Editor's evaluation

This work provides a comprehensive set of convincing biochemical and structural experiments to determine the molecular basis of calcium-sensitive regulation of the guanine exchange factor Ric8A by the neuronal calcium sensor 1 (NCS-1). The Ric-8A/NCS-1 interface is a promising target for modulation of synaptic activity under pathological conditions, and this work will have important implications for scientists interested in G-protein signaling and molecular interactions that contribute to synapse function.

## Introduction

$Ca^{2+}$ is a key signal that regulates multiple biological phenomena ranging from neurotransmission to gene expression. Changes in the concentration of intracellular free $Ca^{2+}$, its locus of action, and the amplitude and duration of $Ca^{2+}$ influx are essential to transmit information through the nervous system. The mechanisms by which these changes can bring about such diverse neural responses rely on the ability of $Ca^{2+}$ sensors to decode $Ca^{2+}$ signals (*McCue et al., 2010*; *Burgoyne et al., 2019*). The neuronal calcium sensor (NCS) family of proteins has evolved to participate in specialized neuronal functions separate from calmodulin, due to their 10-fold higher affinity for $Ca^{2+}$. The most abundant protein of the NCS family is the neuronal calcium sensor 1 (NCS-1), which was first discovered in *Drosophila* and named frequenin (Frq), is N-terminally myristoylated (Myr) and highly conserved from yeast to humans (*Burgoyne et al., 2019*; *Pongs et al., 1993*; *Ames and Lim, 2012*; *Burgoyne and Haynes, 2012*).

Unlike other NCSs, NCS-1 is found outside the nervous system. It does not contain a $Ca^{2+}$/Myr switch, thus being constantly bound to the membrane, and has multiple binding partners (*Burgoyne et al., 2019*; *Ames and Lim, 2012*; *Burgoyne and Haynes, 2012*; *Mansilla et al., 2017*). NCS-1 participates in a wide range of important neuronal functions: it is a regulator of $Ca^{2+}$ channels, exocytosis, synaptogenesis, and axonal growth, affecting higher functions such as learning and memory, neuroprotection, and axonal regeneration (*Burgoyne et al., 2019*; *Burgoyne and Haynes, 2012*; *McFerran et al., 1999*; *Hui and Feng, 2008*; *Weiss et al., 2010*; *Dason et al., 2012*). Furthermore, NCS-1 has been implicated in several pathological processes such as X-linked mental retardation and autism, schizophrenia, and bipolar disorder (*Piton et al., 2008*; *Koh et al., 2003*; *Torres et al., 2009*; *Bahi et al., 2003*). The multifunctionality of NCS-1 relies on its ability to recognize and regulate different and unrelated target proteins: G-protein-coupled receptors (GPCRs) and some of their regulators, $Ca^{2+}$ channels, guanine nucleotide exchange factors (GEF), and kinases, both in a $Ca^{2+}$-dependent or -independent manner (*Burgoyne et al., 2019*).

The structure of NCS-1 consists of two pairs of EF-hand motifs, of which only three are functional: EF-2, EF-3, and EF-4 (*Bourne et al., 2001*). EF-2 and EF-3 can recognize $Mg^{2+}$ as well. It is known that the two $Ca^{2+}$/$Mg^{2+}$ binding sites are structural sites that allow the protein to adopt its tertiary fold. EF-4 has been suggested to be a regulatory $Ca^{2+}$ binding site, able to sense changes in cytosolic calcium levels in neurons (*Burgoyne et al., 2019*; *Aravind et al., 2008*; *Mikhaylova et al., 2009*; *Chandra et al., 2011*; *Tsvetkov et al., 2018*; *da Silva et al., 1995*; *Gifford et al., 2007*).

The structures of several NCS proteins bound to their corresponding targets have shown that these $Ca^{2+}$ sensors use a surface-exposed hydrophobic crevice to recognize their targets, which generally present short helical motifs that bind to the N- or C-terminal part of this large cavity (*Figure 1A*). It has been proposed that the structural determinants of target specificity are based on the shape and size of the hydrophobic crevice. NCS proteins contain a dynamic C-terminal helix (the so-called helix H10) that can insert into the crevice, thus contributing to its shape (*Figure 1A*). Since $Ca^{2+}$ binding promotes structural rearrangements (*Figure 1A*), the occupancy of the three $Ca^{2+}$ binding sites also determines affinity for protein partners. Also, the presence of hydrophilic residues at the border of the crevice contributes to target specificity and they constitute hot spots for interactions with the different targets (*Burgoyne et al., 2019*; *Mansilla et al., 2017*).

Interestingly, NCS-1 is an important regulator of G-protein signaling since it binds to proteins such as GPCRs including the dopamine D2, adenosine 2A, and cannabinoid CB1 receptors (*Kabbani et al., 2002*; *Navarro et al., 2012*; *Pandalaneni et al., 2015*; *Angelats et al., 2018*); and the molecular chaperone and GEF Ric-8A (*Romero-Pozuelo et al., 2014*). Although little is known of the regulatory activity of NCS-1 on GPCR function and the cellular and physiological consequences, *Romero-Pozuelo*

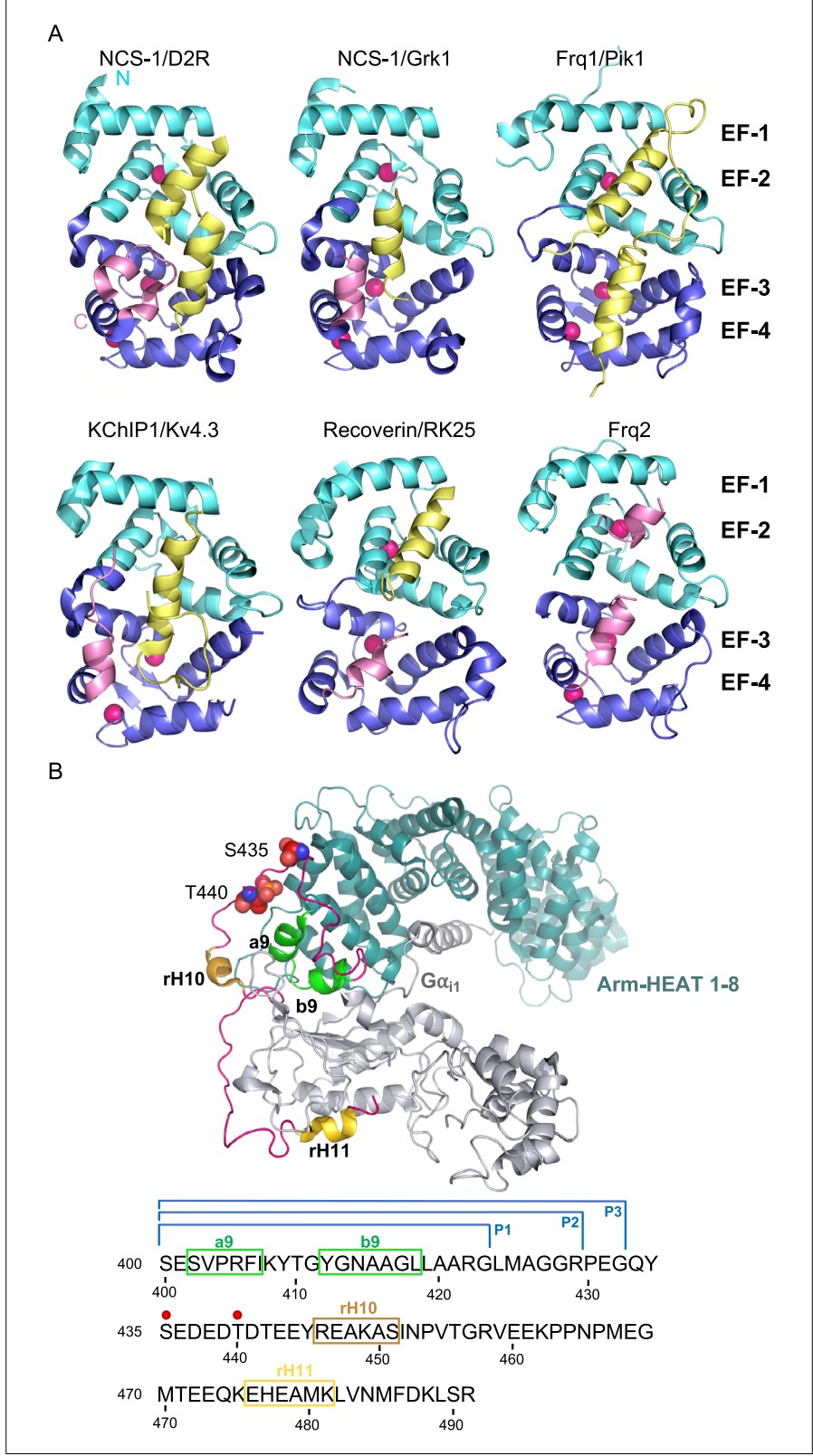

**Figure 1.** The structure of NCS/target complexes. (**A**) Ribbon representation of NCS protein structures bound to their targets. NCS-1/D2R (PDB: 5AER, *Pandalaneni et al., 2015*), NCS-1/Grk1 (PDB: 5AFP, *Pandalaneni et al., 2015*), Frq1/Pik1 (PDB: 2JU0, *Strahl et al., 2007*), KChIP1/Kv4.3 (PDB: 2I2R, *Pioletti et al., 2006*), Recoverin/RK25 (PDB: 2I94, *Ames et al., 2006*), Frq2 (PDB: 4BY4, *Romero-Pozuelo et al., 2014*). The N and C-terminal

*Figure 1 continued on next page*

*Figure 1 continued*

pairs of EF-hands (EF-1–2 and EF-3–4) are shown in cyan and purple respectively. The C-terminal helix H10 is shown in pink and target proteins in yellow. $Ca^{2+}$ is shown as hot pink spheres. (**B**) Top: Cryo-EM structure of the rRic-8A/$G\alpha_{i1}$ complex (PDB: 6UKT, *McClelland et al., 2020*). $G\alpha_{i1}$ is depicted in silver. Ric-8A ARM-HEAT repeats 1–8 in blue and repeat 9 in green and helices H10 (rH10) and H11 (rH11) in orange and gold respectively. The C-terminal coiled regions are shown in magenta. Phosphorylated residues S435 and T440 are depicted as spheres. Bottom: rRic-8A sequence from residue 400 to the end. Helices are squared following the same color code used above. Phosphorylation sites are indicated as red spheres. P1, P2, and P3 brackets indicate the boundaries of the synthesized Ric-8A peptides.

The online version of this article includes the following figure supplement(s) for figure 1:

**Figure supplement 1.** Structural comparison of hNCS-1/Ric-8A-P with other NCS-1 complexes.

---

*et al., 2014*, showed that the NCS-1/Ric-8A complex is implicated in the regulation of synapse number and probability of neurotransmitter release. In fact, the regulation of this protein-protein interaction (PPI) using small-molecule modulators allows synapse function control under pathological conditions. In neurodevelopmental disorders, where synapse number is abnormally high, the inhibition of the NCS-1/Ric-8A complex formation reduces synapse number and improves learning in Fragile X syndrome animal models (*Cogram et al., 2022*; *Mansilla et al., 2017*). In contrast, the stabilization of the PPI prevents synapse loss and the consequent impairment in locomotion in a *Drosophila* model of Alzheimer's disease neurodegeneration at the motor neurons (*Canal-Martín et al., 2019*).

Ric-8A is an ubiquitously expressed cytosolic protein with two main functions: it constitutes a molecular chaperone that allows heterotrimeric Gα subunit biogenesis (*Gabay et al., 2011*) and additionally, works as a guanine exchange factor for $G_i$, $G_q$, and $G_{12/13}$ families (*Tall et al., 2003*; *Chan et al., 2011*; *Van Eps et al., 2015*). Both activities are stimulated by casein kinase II (CK2) phosphorylation (*Papasergi-Scott et al., 2018*). Ric-8A has been shown to regulate asymmetric cell division and is essential for embryonic development (*Miller and Rand, 2000*; *Afshar et al., 2004*; *Couwenbergs et al., 2004*; *Tõnissoo et al., 2010*; *Woodard et al., 2010*). Work in *Drosophila* has shown the relevance of Ric-8A in activating $G_s$ for in vivo synaptogenesis and that this activity is downregulated by NCS-1 (*Romero-Pozuelo et al., 2014*). Recently, the structure of several Ric-8A/Gα or Gα fragment complexes have been solved at atomic level (*Srivastava and Artemyev, 2019*; *McClelland et al., 2020*; *Seven et al., 2020*). These works revealed the structural basis of Ric-8A as a Gα chaperone and GEF and showed how phosphorylation of Ric-8A residues S435 and T440 stabilizes a conformation that is competent for Gα recognition. However, there is scarce information on the molecular function of NCS-1 on Ric-8A activity. Based on genetic studies, it has been proposed that NCS-1 interacts with Ric-8A and prevents the Ric-8A/Gα interaction (*Romero-Pozuelo et al., 2014*). Here, we have combined biochemical, biophysical, and crystallographic studies to reveal the structural determinants of NCS-1/Ric-8A recognition and the mechanism of NCS-1-mediated downregulation of Ric-8A activity. This work shows how NCS-1 and Ric-8A constitute a hub that integrates $Ca^{2+}$, phosphorylation, and G-protein signaling. The emergent picture indicates that Ric-8A activity is under NCS-1 control and that a $Ca^{2+}$ signal triggers the disassembly of the NCS-1/Ric-8A complex, which in turn allows phosphorylation of Ric-8A, which stabilizes the Ric-8A/Gα complex.

## Results

### The NCS-1 interacting region of Ric-8A and the role of $Ca^{2+}$

To identify potential NCS-1 binding regions, we exploited the high-resolution structural information available on Ric-8A and NCS-1. An analysis of the different reported Ric-8A structures was performed to find potential NCS-1 binding regions (*McClelland et al., 2020*; *Zeng et al., 2019*). First, we took into account that NCS protein targets generally employ one or two short helical motifs to recognize the N- or C-terminal pair of EF-hands (*Figure 1A*; *Burgoyne et al., 2019*). Second, we considered that the potential interacting helix or helices may have hydrophobic character, since Ric-8A interacts with NCS-1 through its surface-exposed hydrophobic crevice (*Mansilla et al., 2017*; *Romero-Pozuelo et al., 2014*). Third, NCS-1 and G-proteins compete for Ric-8A binding and thus, could share certain interaction surfaces (*Figure 1B*; *Romero-Pozuelo et al., 2014*). Using these criteria, we evaluated the hydrophobic character of the HEAT repeat 9 of the ARM/HEAT repeat domain of Ric-8A, which

is composed of a two-helix bundle (called a9 and b9), as well as two C-terminal helical motifs (rH10 and rH11), all of them involved in Gα recognition (*Figure 1B*). The marked hydrophobic character of the a9-b9 two-helix bundle led us to hypothesize that a9 and/or b9 helices could be implicated in the interaction with NCS-1.

To test our hypothesis, we carried out the in vitro reconstitution of the protein complex using an unphosphorylated Ric-8A rat variant construct ending at residue 452 (rRic-8A-452), which includes the two-helix bundle a9-b9 and rH10 (*Figure 1B*), and a human NCS-1 deletion construct (NCS-1ΔH10; 100% sequence identity to rat). The NCS-1 C-terminal helix H10 was removed since it works as a built-in competitive inhibitor of the Ric-8A interaction and the affinity for Ric-8A increases 30% in the absence of this helix (*Romero-Pozuelo et al., 2014*). Previous co-immunoprecipitation (co-IP) assays with the NCS-1 *Drosophila* variant suggested that the interaction occurred in the presence of $CaCl_2$, although it was stronger in the presence of EDTA (*Romero-Pozuelo et al., 2014*). To gain further insights into the $Ca^{2+}$ dependency of the complex formation, reconstitution experiments were conducted at different $Ca^{2+}$ concentrations and size exclusion chromatography (SEC) was used to evaluate complex assembly (see Materials and methods section). Assembly (i) was attempted in $Ca^{2+}$-free conditions (2 mM EGTA with or without 1 mM $Mg^{2+}$; see Materials and methods section). No sign of complex assembly was observed (*Figure 2A*). Assembly (ii) was carried out using a $Ca^{2+}$-preloaded NCS-1 protein and maintaining 2 mM $CaCl_2$ concentration in the SEC elution buffer. A very small peak at higher molecular weights was observed and most of the sample eluted as un-complexed species (*Figure 2A*). Therefore, neither a $Ca^{2+}$-free nor a fully $Ca^{2+}$-loaded NCS-1 protein recognized Ric-8A properly, suggesting that $Ca^{2+}$ loading of some, out of the 3 $Ca^{2+}$ binding sites, is required for Ric-8A recognition. To achieve this, $Ca^{2+}$-free EGTA-purified NCS-1 was mixed with Ric-8A at a final EGTA concentration of 0.6 mM, and subsequently dialyzed against a 2 mM $CaCl_2$ buffer (Assembly (iii)). The majority of the protein sample eluted at an apparent molecular mass consistent with the formation of the NCS-1ΔH10/Ric-8A complex, as confirmed by SDS-PAGE gel (*Figure 2A*). Nano-differential scanning fluorimetry (nano-DSF) experiments also corroborated the efficient assembly when moving from EGTA to intermediate $Ca^{2+}$ conditions (*Figure 2B*). In the presence of 0.6 mM EGTA, the denaturation profile of the mixture is consistent with the independent denaturation of both proteins, whose unbound forms have Ti values of 53.5 ± 0.2°C (Ric-8A) and 58.4 ± 0.8°C (NCS-1 EGTA) (*Figure 2B*). In contrast, the thermal transition of Ric-8A is up-shifted by more than 20°C (Ti = 79.7 ± 2.8°C) in the sample dialyzed from EGTA to $Ca^{2+}$ concentrations, due to the complex formation promoted by NCS-1 $Ca^{2+}$ uptake. Moreover, the Ti of the $Ca^{2+}$-bound form of NCS-1 in the sample is 82.8 ± 2.6°C, a value that is significantly lower than that shown when the three $Ca^{2+}$ binding sites are saturated, which is above 90°C (*Figure 2B*).

The Ric-8A construct used in the in vitro reconstitution of the rat NCS-1/Ric-8A complex (rRic-8A-452) includes the two-helix bundle a9-b9 (HEAT repeat 9) and rH10 (*Figure 1B*). To determine whether elements beyond a9-b9 (e.g. rH10 and rH11) are implicated in NCS-1 recognition and also determine if the interaction occurs in the context of the human proteins, a co-immunoprecipitation assay was performed using both full-length human Ric-8A (hRic-8A-FL) and a C-terminally truncated construct lacking rH10 and rH11, and ending at G424 (hRic-8A-424, residues 1–424) (*Figure 2C*). The human Ric-8A sequence is one residue longer than the rat variant due to the insertion of a proline in a loop at position 208. Therefore, G424 in human corresponds to rat G423 (*Figure 1B*). Compared with hRic-8A-FL (*Figure 2C*), hRic-8A-424 has significantly higher affinity for NCS-1, suggesting that the HEAT repeat 9, but not the rH10 or rH11 helices, is implicated in the PPI. This supports a model in which two helices of Ric-8A are bound to NCS-1, similar to the case found for yeast NCS-1 bound to Pik1 (*Figure 1A*; *Strahl et al., 2007*). Also, this model is consistent with a reported crystal structure of *Drosophila* NCS-1 (also known as Frq2) in its apo form (*Romero-Pozuelo et al., 2014*), in which its hydrophobic crevice is occupied by two C-terminal NCS-1 H10 helices, one belonging to the same protein and an additional helix belonging to another molecule of the asymmetric unit, thus mimicking a protein/target complex (*Figure 1A*).

## The crystal structure of hNCS-1 bound to Ric-8A peptides

Attempts to crystallize the assembled in vitro reconstituted NCS-1ΔH10/rRic-8A complex were made with different rRic-8A constructs (ending at different positions between Ric-8A-423 and Ric-8A-452), without success. Therefore, we decided to work with peptides spanning the a9 and b9 helices

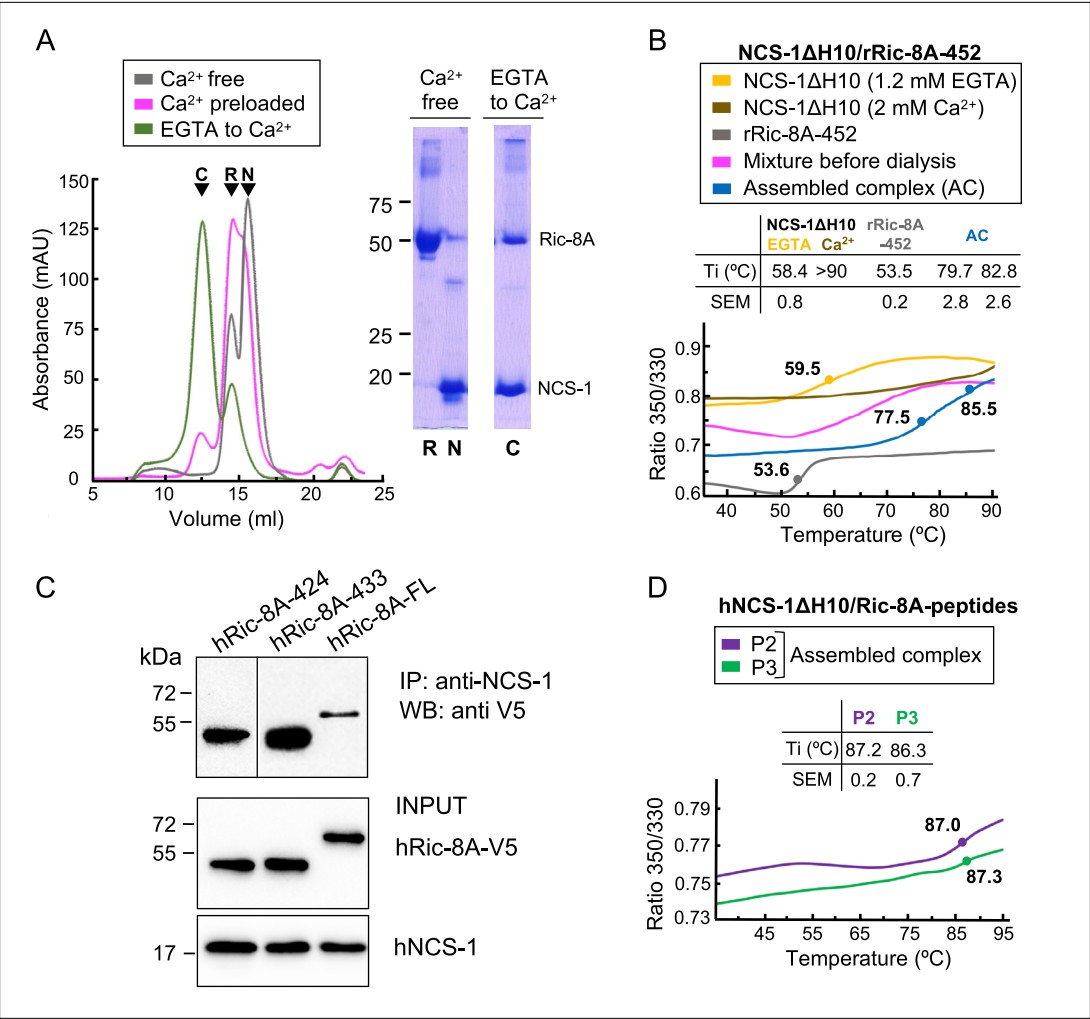

**Figure 2.** The assembly of rat and human NCS-1/Ric-8A complexes. (**A**) $Ca^{2+}$ dependency of the interaction of the rat complex. Size exclusion chromatograms after assemblies: (i) in $Ca^{2+}$-free conditions (gray), (ii) with $Ca^{2+}$-preloaded NCS-1ΔH10 (magenta), and (iii) with a dialysis from EGTA to $Ca^{2+}$ (green). 12% SDS-PAGE gels analyzing elution of NCS-1ΔH10 (N) and rRic-8A-452 (R) after Assembly (i) and the NCS-1/rRic-8A-452 complex (C) after Assembly (iii). (**B**) Representative nano-differential scanning fluorimetry (nano-DSF) curves of the different samples during Assembly (iii). The ratio between the emission fluorescence at 350 nm and 330 nm vs the temperature is shown and dots indicate the inflexion temperature (Ti). Curves corresponding to the EGTA-purified NCS-1 (NCS-1 2 mM EGTA), the fully $Ca^{2+}$ saturated protein (NCS-1 2 mM $Ca^{2+}$) and rRic-8A-452 are shown as references in yellow, brown, and gray, respectively. The mixture of proteins before dialysis (0.6 mM EGTA) and afterward (assembled complex, 2 mM $CaCl_2$) are shown in magenta and blue, respectively. NCS-1 refers to NCS-1ΔH10, while Ric-8A to Ric-8A-452 construct. Three independent measurements were acquired and the inset table summarizes the mean Ti and standard error of the mean (SEM) values of each sample. (**C**) Co-IP protein-protein interaction assay in HEK293 cells of full-length human NCS-1 and V5-tagged hRic-8A constructs: full-length (hRic-8A-FL) and C-terminally truncated hRic-8A-424 (residues 1–424) and hRic-8A-433 (residues 1–433). (**D**) Representative nano-DSF curves of hNCS-1 bound to different Ric-8A peptides. NCS-1 refers to NCS-1ΔH10, P2 and P3 refer to Ric-8A peptides P2 (purple) and P3 (green). Three independent measurements were acquired and the inset table shows the mean Ti and SEM values of each sample.

The online version of this article includes the following source data for figure 2:

**Source data 1.** Original gels, WBs and SEC and nano-DSF raw data.

(*Figure 1B*) to produce a minimal complex (named NCS-1/Ric-8A-P) and perform crystallographic studies. Three different peptides were synthesized starting at residue 400 and ending at positions 423 (P1), 429 (P2), and 432 (P3) (*Figure 1B*). These peptides include a region of Ric-8A that is 100% identical in both the human and rat variants. As indicated above, the human Ric-8A sequence is one residue longer than the rat variant due to the insertion of a proline in a loop at position 208. However, for easier comparison with previous structural studies carried out with the rat variant (*McClelland et al., 2020*; *Zeng et al., 2019*) we have decided to maintain the rat numbering.

The assembly of the minimal complex was performed using conditions similar to those used to form Assembly (iii) as described above. hNCS-1ΔH10 was incubated with the Ric-8A peptides in a 1:10 molar ratio and introducing $Ca^{2+}$ by dialysis, starting at 1.7 mM EGTA and ending with a 0.5 mM $Ca^{2+}$ concentration (see Materials and methods). No crystals were obtained with the shortest peptide (P1). However, crystals were obtained with the complexes assembled with peptides P2 and P3. The crystals obtained with hNCS-1ΔH10/Ric-8A-P2 grew using microseeding techniques in conditions containing 0.5 mM $Ca^{2+}$, 100 mM $Mg^{2+}$, and 100 mM $Na^+$ (see Materials and methods section). hNCS-1ΔH10/ Ric-8A-P3 crystals grew under similar conditions to those found for Ric-8A-P2. In addition, peptide P3 produced crystals in a second condition containing only $Ca^{2+}$ and $Na^+$ (see Materials and methods section). Diffraction data sets were collected at the Spanish ALBA synchrotron (*Table 1*). All crystals belonged to the tetragonal space group $P4_12_12$ and displayed similar cell dimensions. The structure was solved by molecular replacement, using the Ric-8A-P2 data set (Structure 1, *Table 1*).

Crystals of Ric-8A-P2 and -P3 each contain one complex in the asymmetric unit and both the $2F_o$-$F_c$ and the $F_o$-$F_c$ electron density maps clearly indicated the presence of electron density corresponding to two helical segments (R1 and R2) of the peptide that completely occupy the NCS-1 hydrophobic crevice and protrude from its surface (*Figure 3* and *Figure 3—figure supplement 1A*). The quality of the data allowed the unambiguous modeling of Ric-8A-P2 residues 402–429; no density was found for the N-terminal residues S400 and E401 and the density corresponding to residues S402 to R405 was very weak (*Figure 1B* and *Figure 3—figure supplement 1A*). The structures solved with Ric-8A-P3 (Structures 2 and 3, *Table 1*) were virtually identical with the exception that the $Mg^{2+}$ ion is present only in Structures 1 and 2. Compared with Structure 2, the $C_\alpha$ RMSD for the protein and peptide in Structures 1 and 3 were 0.090 and 0.227, and 0.103 and 0.117, respectively. The greatest differences are found between Structures 1 and 2 at the N-terminus of the Ric-8A peptides (*Figure 3—figure supplement 1B*), which make few contacts with NCS-1, and for which the temperature factors are high (*Figure 3—figure supplement 1C*). Although Ric-8A-P3 contains three extra residues at its C-terminus (*Figure 1B*) no electron density was found for the C-terminal residues P430, E431, and G432 suggesting that they are disordered, exposed to the solvent and do not participate in protein-protein recognition (*Figure 3—figure supplement 1A*). In fact, the thermal stability of NCS-1 bound to Ric-8A-P2 or Ric-8A-P3 is very similar (*Figure 2D*). The following discussion focuses on Structure 2, which is determined at the highest resolution and statistical quality (*Table 1*).

The structure of Ric-8A-P bound to NCS-1ΔH10 can be described as a coiled region followed by a short helical motif (R1) that is connected to a long helix (R2) through a loop (*Figure 3A*). The two helical elements are interconnected through the loop by an H-bond and van der Waals contacts (triad 1: I407-T410-A415) (*Figure 3A, B*) and also with helix-helix van der Waals interactions (F406-L418 and triad 2: K408-Y409-N414) (*Figure 3A, B*). Polar contacts with a chloride ion appear to stabilize the conformation of the turn between the two helical segments (*Figure 3A*). A calculation of the surface electrostatic potential of Ric-8A-P shows that the helices are amphipathic and expose positive charges to the solvent, except for the C-terminal tip, which shows a negative potential due to the carboxylic end of the peptide (*Figure 3C*).

The hNCS-1ΔH10/Ric-8A-P contact area is 1140 $Å^2$ (*Krissinel and Henrick, 2007*) and a total of 1469 contacts take place (4.2 Å cutoff distance). All the NCS-1 helices that shape the cavity are implicated in Ric-8A recognition: while R2 helix is recognized by multiple helices, R1 contacts helices H7 and H8 (*Figure 3—figure supplement 2A and B*). A total of nine H-bonds are formed with Ric-8A in the cavity (*Figure 3—figure supplement 2B*). Helix R2 is recognized at the middle (L419), N-(N414) and C-terminus (R429) through H-bonds. The Ric-8A R1-R2 loop also plays a relevant role in the recognition process and Y412 forms two water-mediated H-bonds with residues located at the C-terminal part of the crevice (*Figure 3—figure supplement 2B and C*). NCS-1 residue R148 uses its guanidinium group to establish two direct H-bonds with T410 (R1-R2 loop) and K408 (R2 helix). All NCS-1 residues involved in polar contacts are located at the border of the hydrophobic crevice in which Ric-8A is inserted (*Figure 3—figure supplement 2B*). In addition to the polar contacts, a large number of van der Waals interactions are formed with the bottom and lateral walls of the NCS-1 crevice in which hydrophobic and a good number of aromatic residues are present (*Figure 3—figure supplement 2A*). Several hydrophobic and aromatic Ric-8A residues satisfy these interactions (*Figure 3—figure supplement 2D*).

**Table 1.** Diffraction data collection and refinement statistics of hNCS-1/Ric-8A-P crystals.

| Data collection | Structure 1 | Structure 2 | Structure 3 |
|---|---|---|---|
| PDB code | 8ALH | 8AHY | 8ALM |
| Peptide | P2 | P3 | P3 |
| Ions in solution | $Mg^{2+}$, $Ca^{2+}$, $Na^+$ | $Mg^{2+}$, $Ca^{2+}$, $Na^+$ | $Ca^{2+}$, $Na^+$ |
| Space group | $P4_12_12$ | $P4_12_12$ | $P4_12_12$ |
| Cell dimensions | | | |
| a, b, c (Å) | 56.86, 56.86, 134.61 | 56.64, 56.64, 135.30 | 56.64, 56.64, 134.53 |
| α, β, γ (°) | 90.00, 90.00, 90.00 | 90.00, 90.00, 90.00 | 90.00, 90.00, 90.00 |
| Resolution range (Å) | 52.38–1.86 (1.93-1.86)* [a*, b*=1.846, c*=1.917] | 52.24–1.70 (1.79-1.70)* [a*, b*=1.681, c*=1.891] | 52.20–1.85 (1.94-1.85)* [a*, b*=1.854, c*=1.920] |
| $R_{pim}$ | 0.044 (0.803) | 0.036 (0.659) | 0.028 (0.616) |
| $CC_{1/2}$ | 0.998 (0.445) | 0.997 (0.551) | 0.999 (0.467) |
| I/σI | 16.8 (1.2) | 13.5 (1.4) | 15.2 (1.3) |
| Completeness | | | |
| Spherical (%) | 92.7 (41) | 88.0 (34.1) | 91.2 (36.1) |
| Ellipsoidal (%) | 94.9 (51.5) | 95.9 (65.4) | 94.0 (45.7) |
| Wilson B-factor | 31.12 | 26.90 | 37.80 |
| Multiplicity | 25.2 (26.5) | 25.5 (27.7) | 8.7 (9.8) |
| Refinement | | | |
| Resolution (Å) | 52.38–1.86 | 52.24–1.70 | 52.20–1.85 |
| No. reflections | 18029 | 22008 | 17665 |
| $R_{work}$/$R_{free}$ | 19.58/23.13 (26.31/28.49) | 18.64/20.76 (34.65/45.00) | 20.98/25.25 (36.89/46.21) |
| Asymetric unit content | | | |
| No. atoms | 3454 | 3403 | 3374 |
| Protein (no. residues) | 171 | 171 | 171 |
| Peptide (no. residues) | 28 | 28 | 28 |
| PEG/GOL | 3/1 | 2/1 | 2/2 |
| $Ca^{2+}$/$Cl^-$/$Mg^{2+}$/$Na^+$ ions | 2/1/1/1 | 2/1/1/1 | 2/1/0/2 |
| Water molecules | 141 | 144 | 116 |
| B-factor (Å)$^2$ | 31.27 | 27.28 | 37.18 |
| R.m.s. deviations protein | | | |
| Bond lengths (Å) | 0.44 | 0.36 | 0.50 |
| Bond angles (°) | 0.63 | 0.56 | 0.65 |
| R.m.s. deviations peptide | | | |
| Bond lengths (Å) | 0.45 | 0.57 | 0.33 |
| Bond angles (°) | 0.70 | 0.63 | 0.61 |

*Values in parenthesis are for highest resolution shell.

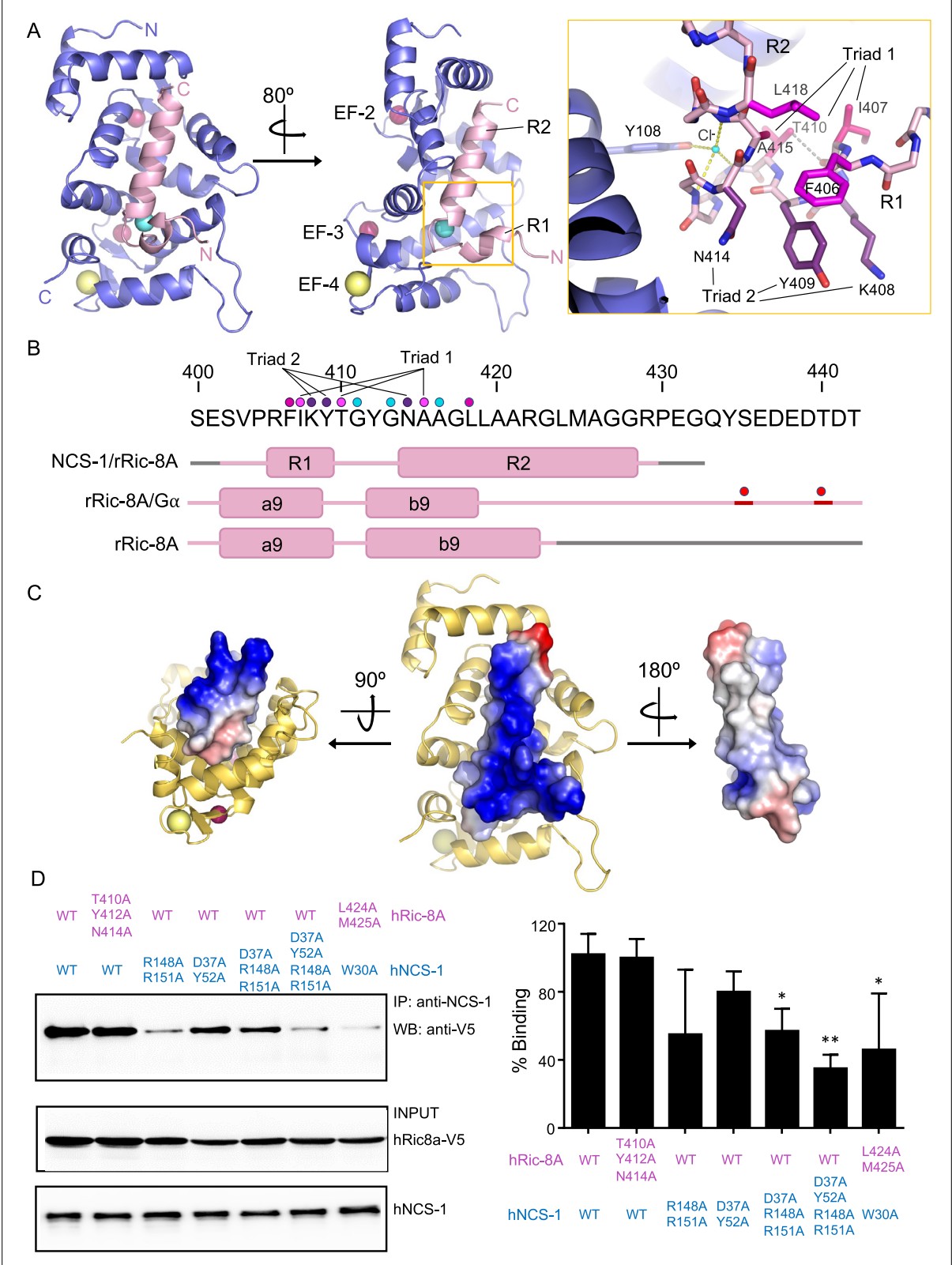

**Figure 3.** The structure of hNCS-1 bound to a Ric-8A peptide. (**A**) Ribbon representation of the hNCS-1ΔH10/Ric-8A-P3 complex. Two views are displayed. The NCS-1 structure is shown in light purple, while Ric-8A-P3 is shown in light pink. The N- and C-termini are indicated. Ca²⁺, Na⁺, and Cl⁻ ions are shown in hot pink, yellow, and cyan, respectively. R1 and R2 helices, and EF-hands 2, 3, and 4 are indicated. The orange square represents a zoomed view of the R1-R2 loop in stick mode, Cl⁻ coordination and H-bonds are displayed as yellow and gray dashes, respectively. Residues

*Figure 3 continued*

participating in R1-R2 contacts are displayed in hot pink (triad 1: I407-T410-A415), magenta (F406-L418), and purple (triad 2: K408-Y409-N414). (**B**) rRic-8A sequence from 400 to 442 residues. The helix boundaries of Ric-8A sequence encompassing a9 and b9 in different structural contexts (NCS-1/Ric-8A-peptide [PDB: 8AHY], Ric-8A/Gα [PDB: 6UKT, *McClelland et al., 2020*] and uncomplexed Ric-8A [PDB: 6NMG, *Zeng et al., 2019*]) are indicated as pink boxes and labeled. Coiled regions are shown in pink. Disordered regions are shown in gray, while phosphorylated sites are shown as red spheres. The interacting residues shown in panel (**A**) are indicated with dots in the same color code as in **A**. (**C**) Electrostatic surface potential of rRic-8A-P3. NCS-1 is shown as yellow ribbons. Positive and negative potentials are represented in blue and red, respectively. On the right, the Ric-8A region that faces and contacts NCS-1 is shown with NCS-1 removed for proper visualization. (**D**) Representative co-immunoprecipitation assay in HEK293 cells transfected with full-length hNCS-1 and V5-tagged hRic-8A mutants. Mutations on NCS-1 and Ric-8A are indicated in blue and pink, respectively. The numbering of the rat Ric-8A sequence has been maintained for proper comparison with A and B. Quantifications of each lane from three independent experiments (mean ± SD) are shown on the right. Mean differences were analyzed by two-tailed, paired Student's t-test, comparing with wild-type NCS-1 and Ric-8A. **p=0.01; *p=0.05.

The online version of this article includes the following source data and figure supplement(s) for figure 3:

**Source data 1.** Original WBs.

**Figure supplement 1.** Structure resolution of hRic-8A peptides bound to NCS-1.

**Figure supplement 2.** The hNCS-1/Ric-8A-P protein-protein interface.

Using Ric-8A peptides of different lengths to generate crystal structures, together with PPI assays, it has been possible to define the region of Ric-8A that is necessary and sufficient for NCS-1 recognition and which is conserved in rat and human protein sequences. The addition of three extra residues (peptide P3 vs P2) may allow better folding of the R2 helix. In fact, a cell-based PPI assay with an hRic-8A deletion mutant (hRic-8A-433) ending like rRic-8A-P3 at G432 shows the strongest interaction with full-length hNCS-1 (*Figure 2C*). The crystal structure shows that all Ric-8A-P modeled residues except the N-terminal S402, and the residues P404, R405, and F406 (which belong to helix R1 and are exposed to the solvent, opposite to the face that contacts NCS-1) (*Figure 3—figure supplement 1A*, *Figure 3B*, *Figure 3—figure supplement 2B-D*) are involved in the interaction. The long R2 helix highly contributes to the protein-protein contact area with multiple van der Waals contacts and four H-bonds. Shorter regions such as the R1-R2 loop, only formed by residues T410, G411, Y412, and G413, are also implicated in the PPI, contributing with three hydrogen bonds (*Figure 3—figure supplement 2*). In fact, these four residues account for 28% of total contacts between hNCS-1 and Ric-8A-P.

To further validate the PPI interface and the relevance of the interactions observed in the presented crystal structures, several Ric-8A and NCS-1 full-length human mutant proteins were generated (*Figure 3* and *Figure 3—figure supplement 2B*) and co-immunoprecipitations were conducted to analyze their impact on the recognition of the proteins (*Figure 3D*). Most of the strong and directional interactions, hydrogen bonds, between NCS-1 and Ric-8A peptides are mediated by NCS-1 side chains oxygens and Ric-8A main chain carbonyl oxygens (*Figure 3—figure*

**Table 2.** Residues mutated to alanine to validate the NCS-1/Ric-8A PPI interface.

| Residue | Position and interacting residues |
|---|---|
| NCS-1 D37 | Upper part of the crevice. Interacts with Ric-8A R429, which is located at the C-terminal end of R2 helix |
| NCS-1 Y52 | Middle of the crevice. Recognizes Ric-8A L419, which is found at the middle of R2 helix |
| NCS-1 R148 | Bottom of the crevice. Interacts with Ric-8A K408 (N-terminus of R1 helix) and Ric-8A T410 (R1-R2 loop) |
| NCS-1 R151 | Bottom of the crevice. Interacts with Ric-8A K408 (N-terminus of R1 helix) and Ric-8A Y412 (R1-R2 loop, water-mediated H-bond) |
| Ric-8A T410, Y412, N414 | R1-R2 loop. Mediate several water-mediated H-bonds and van der Waals contacts with the bottom surface of NCS-1 crevice |
| NCS-1 W30A | Upper part of the cavity. Important in the recognition of R2 helix. Establish van der Waals interactions with residues such as Ric-8A L424 and M425 |
| Ric-8A L424 and M425 | C-terminal part of helix R2. Interact with NCS-1 W30 and the hydrophobic environment that surrounds these residues |

*supplement 2B*). Therefore, we introduced mutations in NCS-1 and tested the interaction with wild-type hRic-8A. To disrupt H-bonds found in the upper, middle, and/or bottom of the NCS-1 cavity, two double mutants (NCS-1 D37A, Y52A and NCS-1 R148A, R151A) a triple mutant (NCS-1 D37A, R148A, R151A) and a quadruple mutant including the previous double mutants (NCS-1 D37A, Y52A, R148A, R151A) were generated. The location and interacting residues of mutated amino acids are shown in *Figure 3—figure supplement 2B* and summarized in *Table 2*. The co-immunoprecipitation assays show that mutations R148A, R151A in NCS-1 do not alter Ric-8A binding, despite three H-bonds are lost at the bottom of the crevice. When the recognition of Ric-8A R2 helix is affected at the middle and C-terminus of NCS-1 crevice (Y52A and D37A, respectively), differences in binding are not significant either. The combination of mutations at the top and bottom of NCS-1 crevice (triple mutant; D37A, R148A, R151A), which would affect the recognition of both Ric-8A helices and the R1-R2 loop, has a statistically significant impact on the interaction (40% binding loss). As expected, the quadruple mutant (D37A, Y52A, R148A, R151A) shows the deepest impact on the interaction (more than 60% binding loss), since H-bonds are disrupted at the top, middle, and bottom of the PPI. To further understand the relevance of the R1-R2 loop (*Table 2* and *Figure 3—figure supplement 2C–D*), a Ric-8A triple mutant, T410A, Y412A, N414A (which would correspond to T411A, Y413A, N415A in the human protein), was tested in the PPI binding assay showing no changes in NCS-1 binding. This is in agreement with the previous NCS-1 double mutant designed in that area (R148A, R151A). Finally, we designed mutants in NCS-1 and Ric-8A to test the relevance of the interactions of the C-terminal half of Ric-8A R2 helix, which acquires a extended coiled structure in the presence of Gα (*McClelland et al., 2020*; *Seven et al., 2020*) and becomes helical for NCS-1 recognition. Mutations include NCS-1 W30 (W30A) and two Ric-8A residues close to NCS-1 W30, L424, and M425 (L425 and M426 in the human protein) (*Table 2* and *Figure 3—figure supplement 2A and D*). Despite these residues only mediate long-range hydrophobic contacts, and no H-bond is affected, the impact on the PPI is relevant and binding is reduced more than 50%. The effect is similar to that found when disrupting H-bonds all along the PPI (NCS-1 quadruple mutant).

## The $Ca^{2+}$ binding sites of hNCS-1 in complex with Ric-8A-P

The analysis of the structure of the hNCS-1ΔH10/Ric-8A-P complexes presented in this work, together with electron density map calculations, indicate that the three $Ca^{2+}$ binding sites, EF-2, EF-3, and EF-4, are occupied, showing a pentagonal-bipyramidal coordination (*Figure 4A*). However, the in vitro reconstitution assays presented above (*Figure 2A*) show that the fully $Ca^{2+}$ saturated protein does not efficiently generate the NCS-1/Ric-8A complex. Therefore, some out of the three sites are not occupied with $Ca^{2+}$. NCS-1 $Ca^{2+}$ occupancy has been addressed previously with the calculation of anomalous difference maps since $Ca^{2+}$ but not $Mg^{2+}$ or $Na^+$ shows anomalous signal at typical protein-diffraction wavelengths (*Mansilla et al., 2017*). In fact, at 0.979 Å wavelength, the anomalous scattering coefficients (f") for $Ca^{2+}$, $Na^+$, and $Mg^{2+}$ are 0.616, 0.049, and 0.073 electrons, respectively. Therefore, to identify the $Ca^{2+}$ ions bound to the EF-hands, anomalous difference maps were calculated as a 10-fold increased anomalous signal is expected if $Ca^{2+}$ is present. We used the best quality data set (Structure 2), in which $Ca^{2+}$, $Na^+$, and $Mg^{2+}$ were present in the crystallization solution. The anomalous difference map shows 6σ peaks at EF-2 and EF-3 metal sites while no signal is observed in EF-4, indicating unambiguously the presence of $Ca^{2+}$ at sites EF-2 and EF-3 (*Figure 4A*). EF-4 shows a residual anomalous signal when reducing the σ level, which could indicate a low occupancy of $Ca^{2+}$ at EF-4, consistent with the observation that a fully saturated NCS-1 does not recognize Ric-8A properly (*Figure 2A*). We have not modeled $Mg^{2+}$ at any of the metal binding sites since it has been reported that EF-4 is unable to bind this metal (*Aravind et al., 2008*). Furthermore, the metal-oxygen distances observed (2.3–2.4 Å) are higher than those characteristic of $Mg^{2+}$ coordination (2.1 Å) (*Harding, 2001*; *Harding, 2002*). We did observe a $Mg^{2+}$ ion at the surface of the NCS-1, close to EF-3 (*Figure 4A*) and located at a special position where atoms from two symmetry-related molecules participate in the coordination sphere. In this case, $Mg^{2+}$ exhibits typical octahedral coordination and metal-oxygen distances (2.1 Å), as well as the absence of an anomalous signal. Taking these observations together, a $Na^+$ ion was modeled at EF-4 (*Figure 4A*), since $Na^+$ was present in the crystallization solution, does not scatter X-rays anomalously, and is structurally undistinguishable from $Ca^{2+}$. Their similar ionic radii allow equivalent heptahedral geometry, distances, and angles (*Harding, 2002*; *Groom et al., 2016*).

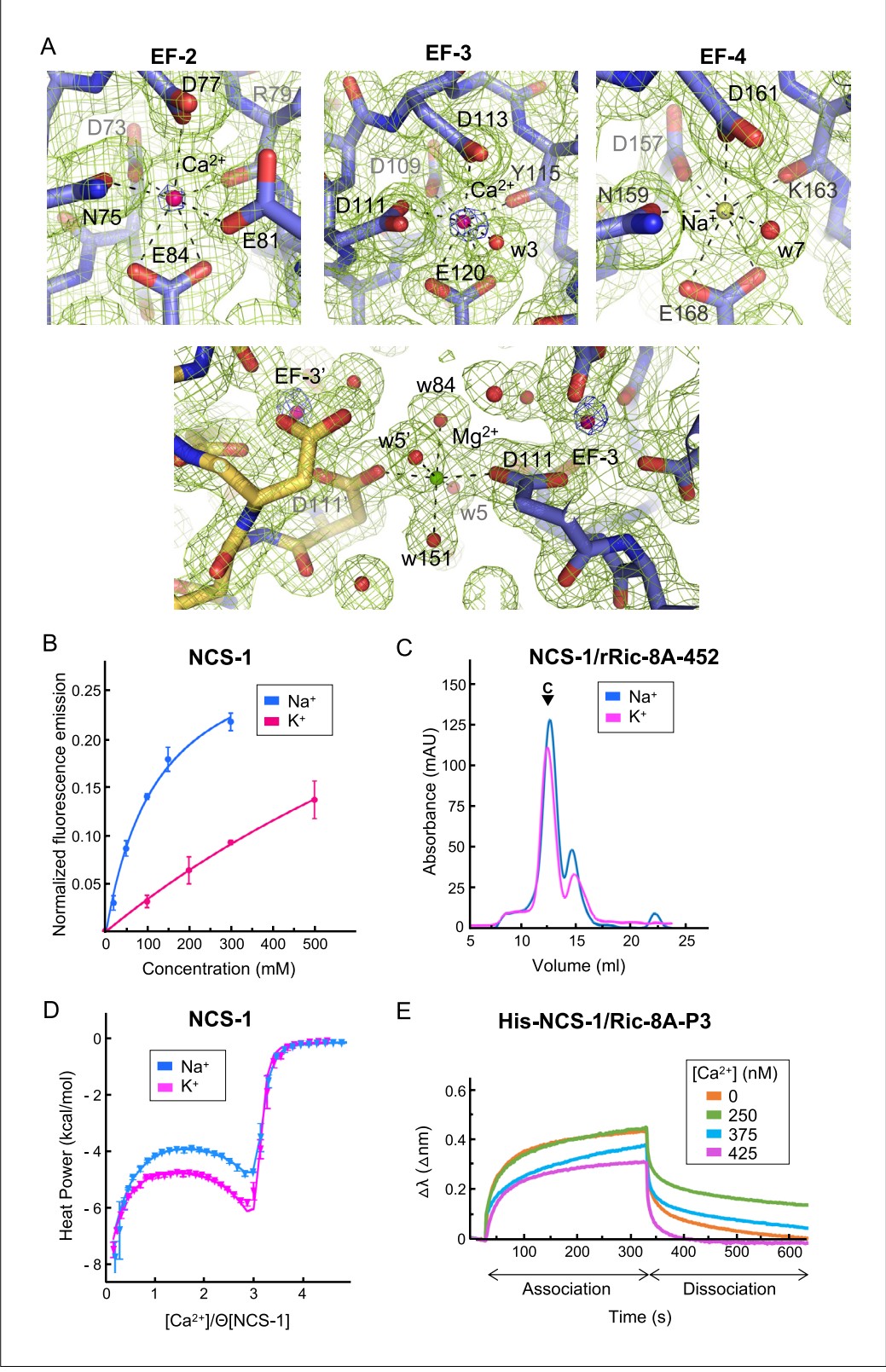

**Figure 4.** NCS-1 $Ca^{2+}$ binding sites. (**A**) Identification of $Ca^{2+}$, $Mg^{2+}$, and $Na^+$ ions in the hNCS-1ΔH10/Ric-8A-P3 complex (Structure 2, see *Table 1*). Top: Electron density at EF-hands EF-2, -3 and -4. The $2F_o$-$F_c$ electron density map (contoured at 1.0σ) and the anomalous difference map (contoured at 6.0σ) are shown in green and blue, respectively. NCS-1 is shown in stick mode (light purple), $Ca^{2+}$ and $Na^+$ ions as hot pink and yellow spheres,

*Figure 4 continued on next page*

*Figure 4 continued*

respectively, and water molecules (w) as red spheres. Bottom: The $Mg^{2+}$ ion (green sphere) found in Structures 1 and 2 (see *Table 1*). NCS-1 symmetry-related molecule is depicted in yellow. (**B**) The binding of $Na^+$ to hNCS-1 in solution. Representation of the normalized fluorescence emission (mean ± standard error of the mean [SEM]; n=3) of full-length hNCS-1 at increasing concentrations of NaCl or KCl. The curves are the least squares fitting of the experimental data to a 1:1 stoichiometry equilibrium. $Na^+$ and $K^+$ titrations are shown in blue and magenta, respectively. (**C**) Assembly of the NCS-1ΔH10/rRic-8A-452 complex in the presence of 200 mM $Na^+$ (blue) or $K^+$ (magenta). Size exclusion chromatograms indicating the elution of the assembled complexes (**C**). (**D**) Isothermal titration calorimetry (ITC) binding isotherm at 25°C for $Ca^{2+}$ to NCS-1 in 20 mM Tris pH 7.9 supplemented with 150 mM $Na^+$ (blue) or 150 mM $K^+$ (magenta). Solid lines show the best fits of the titration data in terms of a three-site sequential binding model using the thermodynamic parameters shown in *Table 4*. Θ is the fraction of sites available for each class of $Ca^{2+}$ sites. (**E**) The binding of full-length His-NCS-1 to Ric-8A-P3 peptide at increasing $Ca^{2+}$ concentrations. Representative biolayer interferometry (BLI) sensograms showing association and dissociation of Ric-8A-P3 over the time. Data are represented as the wavelength shift, $\triangle\lambda$ (nm), during the association and dissociation phases (s).

The online version of this article includes the following source data and figure supplement(s) for figure 4:

**Source data 1.** Raw chromatograms, nano-DSF and Bli data.

**Figure supplement 1.** Isothermal titration calorimetry.

**Figure supplement 2.** Biolayer interferometry (BLI) control experiments.

EF-hand containing proteins, for example, parvalbumin, are able to bind $Na^+$ at their $Ca^{2+}$ binding sites (*Grandjean et al., 1977*). Using tryptophan emission fluorescence experiments, we studied the ability of full-length hNCS-1 to bind $Na^+$ in solution (*Figure 4B*). As a control, we also studied the binding to $K^+$. Our data show that an increase in $Na^+$ concentration produces an enhancement of the emission fluorescence intensity that achieves saturation at 300 mM NaCl. Considering a 1:1 equilibrium, it is possible to estimate the apparent dissociation constant from these data, which is 123.4±26.6 mM (*Mansilla et al., 2017*). Compared with $Na^+$, the addition of $K^+$ promotes smaller changes in the emission intensity, which varied linearly with the cation concentration up to 500 mM KCl, with no sign of saturation. This suggests that NCS-1 is unable to bind $K^+$ at physiologically relevant concentrations (*Figure 4B*). Therefore, the tryptophan emission fluorescence experiments support the binding of $Na^+$ to NCS-1. The changes observed upon $Na^+$ binding may reflect variations in the environment of W103 (EF-3 helix H6), which is the tryptophan residue closer to the EF-4 $Na^+$ binding site. In fact, the EF-4 helix H9 is connected to EF-3 helix H6 through multiple hydrophobic contacts (*Figure 3—figure supplement 2*), so that the structural rearrangement that occurs upon $Na^+$ binding at EF-4 could be transmitted to W103. Indeed, EF-hand motifs always occur in pairs and upon $Ca^{2+}$ binding, EF-hands communicate with each other through the $Ca^{2+}$ binding loops and helix-helix contacts, in order to orchestrate structural rearrangement and transmit the $Ca^{2+}$ signal (*Grabarek, 2006*). Here, we show that $Na^+$ binding to a $Ca^{2+}$ binding loop is also transmitted between EF-hands.

To determine whether the binding of $Na^+$ to EF-4 is relevant for NCS-1/Ric-8A assembly, we compared the in vitro assembly of rat Ric-8A-452 and NCS-1ΔH10 in the presence of 200 mM NaCl or KCl and 2 mM $Ca^{2+}$ (i.e. the so-called Assembly (iii) in the presence of NaCl or KCl). The formation of the complex is achieved in the presence of either ion (*Figure 4C*). Therefore, as suggested by the $Ca^{2+}$ titration experiments (*Figure 2A*) only binding of $Ca^{2+}$ to the structural sites EF-2 and EF-3 (*Aravind et al., 2008*) triggers the conformational rearrangement needed for Ric-8A recognition, and $Na^+$ occupation of EF-4 does not contribute to NCS-1/Ric-8A complex formation. Therefore, Ric-8A can bind to NCS-1 species with a $Ca^{2+}$-free EF-4 site, and in this situation, and at the 100 mM $Na^+$ concentrations used in the crystallization conditions, the site is occupied with $Na^+$.

Given that NCS-1 EF-4 binds $Ca^{2+}$ (*Bourne et al., 2001*) but also $Na^+$, we used isothermal titration calorimetry (ITC) to determine whether

**Table 3.** The $Ca^{2+}$-dependent affinity of full-length NCS-1 for Ric-8A-P3 peptide.

Calculated apparent $K_d$ and standard error of the mean (SEM) using biolayer interferometry. Three independent experiments were performed at each [$Ca^{2+}$].

| | [$Ca^{2+}$] (nM) | | | |
|---|---|---|---|---|
| | **0** | **250** | **375** | **425** |
| $K_d$ (μM) | 140 | 344 | 381 | 98620 |
| SEM | 29 | 14 | 18 | 16310 |

**Table 4.** Thermodynamic parameters of $Ca^{2+}$ binding to full-length hNCS-1 in the presence of $K^+$ or $Na^+$.

| C (150 mM) | $K_{d1}$ (nM) | $\Delta H_1$ (kcal/mol) | $K_{d2}$ (nM) | $\Delta H_2$ (kcal/mol) | $K_{d3}$ (nM) | $\Delta H_3$ (kcal/mol) |
|---|---|---|---|---|---|---|
| $Na^+$ | 265±6 | −7.7±0.1 | 758±16 | 3.0±0.1 | 379±17 | −9.1±0.3 |
| $K^+$ | 165.6±0.3 | −7.66±0.01 | 362.3±0.6 | 1.00±0.01 | 253±1 | −9.44±0.01 |

Subscripts 1, 2, and 3 correspond to sites 1, 2, and 3, respectively.

binding of $Na^+$ to EF-4 affects the $Ca^{2+}$ binding properties of NCS-1. We compared the binding isotherms obtained in the presence of 150 mM NaCl or KCl (control). In both media the binding data can be fit to a sequential binding model assuming three available sites (*Figure 4D*), the first and last binding steps being enthalpy-driven and the second step entropy-driven. As shown in *Figure 4—figure supplement 1*, filling of the third site is significantly delayed compared to the occupation of the other two sites. Since the stoichiometric binding model makes no distinction as to which sites are saturated, but only on the total number of saturated sites, establishing a correlation between the ITC-derived parameters and filling of specific sites is not possible. However, a previous NMR-based study showed that $Ca^{2+}$ sequentially binds to EF-2, EF-3, and finally EF-4, based in the microscopic binding constant measured for each site (*Aravind et al., 2008*; *Mikhaylova et al., 2009*; *Chandra et al., 2011*; *Tsvetkov et al., 2018*). This observation is also consistent with our crystal structure, where $Ca^{2+}$ is found only in EF-2 and EF-3 sites (*Figure 4A*). Interestingly, a direct inspection of the isotherms suggests that the interaction with $Ca^{2+}$ depends on the monovalent cation present in the media. In particular, we observe that the dissociation constants of the first and last steps (likely involving sites EF-2 and EF-4) increase by about 1.5 times in the presence of $Na^+$, while that of the second step doubles and its enthalpy change triples. In conclusion, our data suggest that $Na^+$ binding modulates the affinity of NCS-1 for $Ca^{2+}$, the influence likely extending to more than one site due to communication among them (*Gifford et al., 2007*). However, the possibility that $Na^+$ could bind to more than one unoccupied $Ca^{2+}$ site cannot be completely discarded.

Taking into account the $Ca^{2+}$ affinity measured for the regulatory EF-hand (Table 4) and the fact that the $Ca^{2+}$ saturated protein does not recognize NCS-1, we performed biolayer interferometry (BLI) experiments to study the $Ca^{2+}$-dependent affinity of NCS-1 for Ric-8A, using the full-length NCS-1 and the Ric-8A-P3 peptide (see Materials and methods section, *Figure 4E*, *Figure 4—figure supplement 2* and *Table 3*). Our data show that when the regulatory EF-4 $Ca^{2+}$ binding site is empty (0 nM $Ca^{2+}$ and 125 mM $Na^+$ experimental conditions), the apparent dissociation constant of the NCS-1/Ric-8A-P3 complex is in the hundreds of the µM range. However, at high cellular $Ca^{2+}$ concentrations (250 nM), the affinity is decreased 1.5 times. At 425 nM $Ca^{2+}$, when $Ca^{2+}$ concentrations are above the $K_d$ of the regulatory EF-4 $Ca^{2+}$ binding site (see *Table 4*, site 3), NCS-1 affinity for Ric-8A-P3 is greatly reduced to the mM range.

## Ric-8A phosphorylation in the context of the NCS-1/Ric-8A complex

Phosphorylation of rRic-8A at S435 and T440 residues promotes Gα subunits binding, since the interaction of the phosphorylated residues with a basic groove found at the ARM-HEAT repeat domain of Ric-8A creates a structural platform that allows Gα recognition (*Figure 1B*; *Srivastava and Artemyev, 2019*; *McClelland et al., 2020*; *Seven et al., 2020*). Here, we have shown that in vitro, the NCS-1ΔH10/rRic-8A complex assembly occurs with unphosphorylated rRic-8A. Thus, we have generated the corresponding non-phosphorylatable human full-length Ric-8A mutant (S436A, T441A; Ric-8A-P-Mut) and tested the binding to full-length hNCS-1 in a cell-based PPI assay (*Figure 5A*). The co-immunoprecipitation of the protein complex shows that the non-phosphorylatable version of hRic-8A retains its ability to interact with hNCS-1. Interestingly, the binding is increased with respect to the wild-type version, which would suggest that Ric-8A is partly phosphorylated in vivo and this might hinder the interaction with NCS-1.

To show whether NCS-1 has an influence on Ric-8A phosphorylation, we attempted to phosphorylate the NCS-1ΔH10/rRic-8A-452 complex using CK2 following a protocol similar to that described by *McClelland et al., 2020*. A control experiment was performed with the uncomplexed rRic-8A-452 protein. Protein phosphorylation was evaluated by anion exchange chromatography after incubation

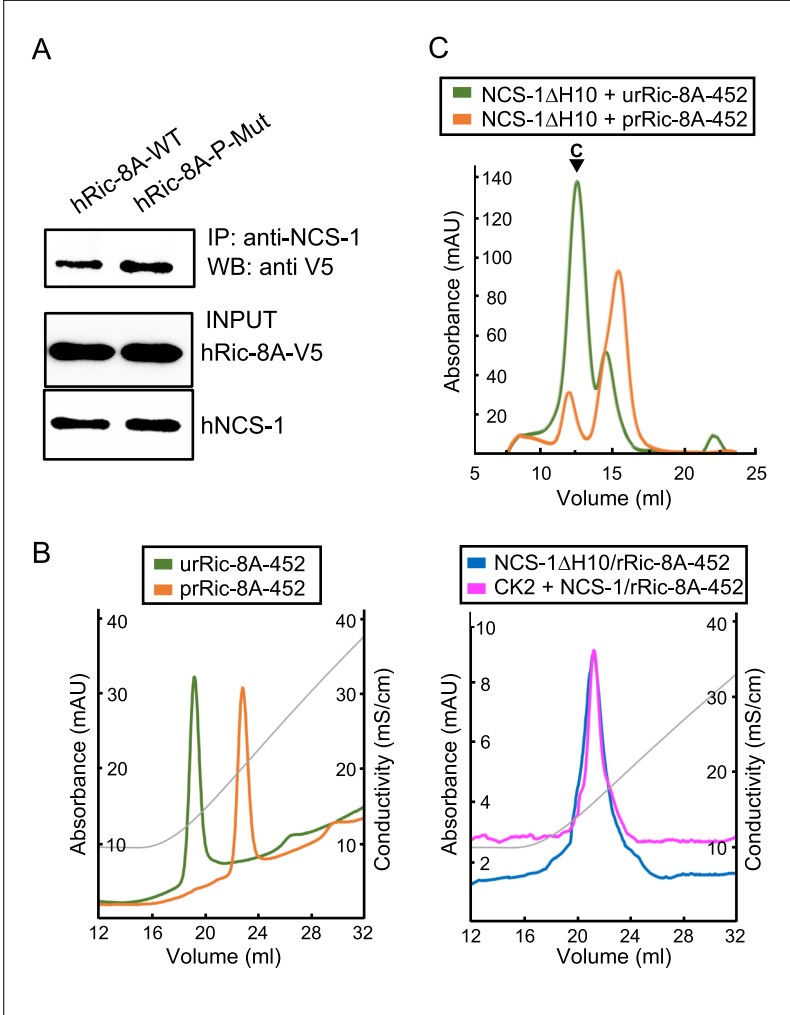

**Figure 5.** Ric-8A phosphorylation in the context of the NCS-1/Ric-8A complex. (**A**) Co-IP protein-protein interaction assay of hNCS-1 and V5-tagged full-length hRic-8A wild-type (WT) (hRic-8A-WT) and a non-phosphorylatable mutant (Ric-8A-P-Mut; S436A, T441A) in HEK293 cells. (**B**) Anionic exchange chromatograms of casein kinase II (CK2)-treated samples eluted in a salt gradient. On the left, phosphorylated and unphosphorylated rRic-8A-452 prRic-8A-452 (orange) and urRic-8A-452 (green), respectively. On the right, CK2 treated (pink) or untreated (blue) NCS-1ΔH10/rRic-8A samples. Conductivity (mS/cm) is shown as gray lines. (**C**) Size exclusion chromatograms of the resulting samples after the assembly of NCS-1ΔH10 with unphosphorylated (green) and phosphorylated (orange) rRic-8A-452. C stands for assembled complex.

The online version of this article includes the following source data and figure supplement(s) for figure 5:

**Source data 1.** Original WBs and raw chromatogram data.

**Figure supplement 1.** Analysis of phosphorylated rRic-8A-452 by mass spectrometry.

with CK2. The phosphorylated species elute at higher salt concentration since their anionic character is increased with the incorporation of the phosphate groups. As shown in *Figure 5B*, while uncomplexed rRic-8A-452 is phosphorylated after CK2 treatment, rRic-8A-452 is not phosphorylated when complexed with NCS-1, since the anionic exchange elution profile of the NCS-1ΔH10/rRic-8A-452 complex is the same regardless of CK2 treatment. To further verify the phosphorylation state of Ric-8A in the CK2-treated samples, we used LC-MS/MS with the aim of identifying phosphopeptides from a trypsin-digested sample (see Materials and methods). While phosphopeptides were found in the rRic-8A-452 control sample (*Figure 5—figure supplement 1*) no phosphopeptides were found in NCS-1ΔH10/rRic-8A-452, even though a phosphopeptide enrichment protocol was performed, supporting the hypothesis that the binding of NCS-1 precludes CK2-mediated phosphorylation of Ric-8A.

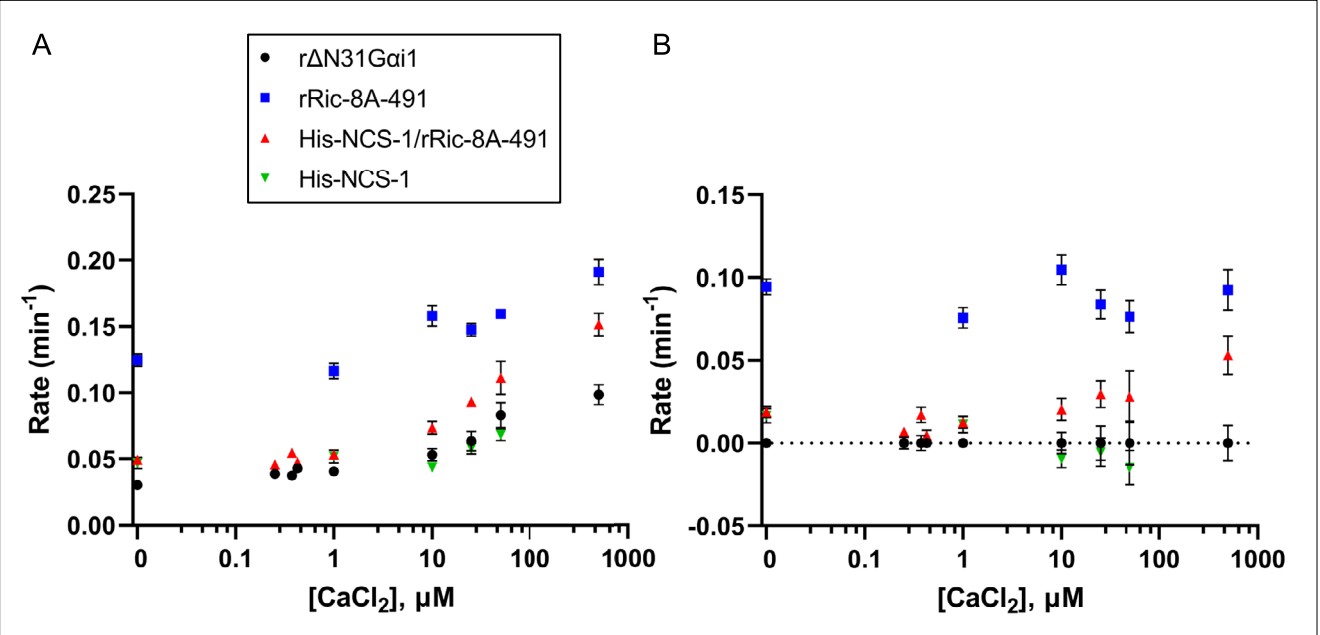

**Figure 6.** Effect of full-length NCS-1 on guanine nucleotide exchange factor (GEF) activity of rRic-8A-491. (**A**) GTP binding rates were measured by following the increase in $r\Delta N31G\alpha_{i1}$ tryptophan fluorescence following addition of 10 μM GTPγS. Prior to GTPγS addition, $r\Delta N31G\alpha_{i1}$ (1 μM final concentration) was incubated with either rRic-8A-491 (0.5 μM final concentration), His-NCS-1/rRic-8A-491 complex (0.5 μM final concentration), or His-NCS-1 (0.5 μM final concentration) and 0–500 μM $CaCl_2$ for 5 min before addition of GTPγS (buffer: 50 mM HEPES pH 8, 200 mM NaCl, 2 mM $MgCl_2$, and 1 mM TCEP). In all cases means and standard error of the mean (SEM) are reported for a minimum of six experimental replicates. (**B**) GTP binding rates of rRic-8A-491, His-NCS-1/rRic-8A-491, and His-NCS-1 after subtraction of intrinsic binding rates of $r\Delta N31G\alpha_{i1}$ at each corresponding $CaCl_2$ concentration.

The online version of this article includes the following figure supplement(s) for figure 6:

**Figure supplement 1.** GTPγS binding progress curves.

**Figure supplement 2.** GTP binding rates vs $CaCl_2$ concentration. GTP binding rates of $r\Delta N31G\alpha_{i1}$ vs $CaCl_2$ concentration, after subtraction of the intrinsic rate at 0 μM $CaCl_2$ (black spheres).

Finally, we asked whether phosphorylated Ric-8A can interact with NCS-1. An assembly experiment was performed similarly to that described for the unphosphorylated protein (control). Our data show that phosphorylated rRic-8A-452 does not efficiently interact with NCS-1ΔH10 (**Figure 5C**).

## NCS-1/Ric-8A nucleotide exchange functional assays

To directly measure the effect of NCS-1 on Ric-8A GEF activity, nucleotide exchange assays were performed in the presence of increasing concentrations of $CaCl_2$. With the aim to approximate physiological conditions, full-length NCS-1 and a more native-like rRic-8A construct (residues 1–491) were used. The change in tryptophan fluorescence of rat $\Delta N31G\alpha_{i1}$ upon exchange of GDP for GTPγS was measured for the intrinsic exchange activity of $\Delta N31G\alpha_{i1}$ and for exchange activity following incubation with either rRic-8A-491, His-NCS-1/rRic-8A-491, or His-NCS-1 in the presence of 0–500 μM $CaCl_2$ (**Figure 6** and **Figure 6—figure supplement 1**). Nucleotide exchange rates were determined by fitting data to a single exponential curve, or in the case of rRic-8A-491 not complexed with NCS-1, a double exponential curve in which the slower of these two rates (**Figure 6**) corresponds to Ric-8A-491-catalyzed nucleotide exchange. The fast phase corresponds to GTPγS binding to an intermediary complex of Ric-8A with nucleotide-free $\Delta N31G\alpha_{i1}$ that is generated after GDP is released from $\Delta N31G\alpha_{i1}$ upon binding to Ric-8A during incubation of the two proteins. This intermediary complex was not observed in assays using the NCS-1/Ric-8A complex, possibly because Ric-8A is partially kinetically trapped in that complex and unable to interact with $\Delta N31G\alpha_{i1}$. The extensive conformational changes that must occur in order to disengage the a9-b9 HEAT repeat helices of Ric-8A from NCS-1 and, subsequently, rearrange to quickly accommodate binding to $G\alpha_{i1}$, likely imposes a substantial kinetic barrier. Also, in vitro and in the absence of other NCS-1 interacting targets, NCS-1/Ric-8A

complex formation is thermodynamically favored since the complete NCS-1 hydrophobic crevice is occluded from the solvent.

In the absence of $Ca^{2+}$, Ric-8A-491 increases the rate of nucleotide exchange at $\Delta N31G\alpha_{i1}$ almost fourfold over the intrinsic rate, whereas His-NCS-1/Ric-8A-491 does not significantly affect the exchange rate (*Figure 6*). We found that $Ca^{2+}$ increases the intrinsic and Ric-8A-catalyzed rates of nucleotide exchange at $\Delta N31G\alpha_{i1}$ in equal measure (*Figure 6A*). However, after correcting for this effect, it is evident that $Ca^{2+}$ produces a concentration-dependent enhancement of the exchange rate in the presence of His-NCS-1/Ric-8A-491 (*Figure 6B*). At 25 µM $CaCl_2$, ~35% of the nucleotide exchange rate is restored to levels similar to that observed in the presence of Ric-8A-491 alone (*Figure 6B*). Thus, NCS-1 complexed to Ric-8A effectively inhibits the GEF activity of Ric-8A whereas addition of $CaCl_2$ restores GEF activity. From these data, it is possible to calculate the apparent activation constant for $Ca^{2+}$-induced enhancement of the intrinsic and Ric-8A-catalyzed nucleotide exchange rates at $\Delta N31G\alpha_{i1}$ ($K_{a-EXC}$=29 ± 8 µM, *Figure 6—figure supplement 2*). In addition, it is possible to estimate a $Ca^{2+}$-induced apparent activation constant for its action on the NCS-1/Ric-8A complex, which disassembles upon $Ca^{2+}$ binding to NCS-1 EF-4, enabling Ric-8A to catalyze $G\alpha_{i1}$ nucleotide exchange ($K_{a-NCS-1}$=61 ± 35 µM). It is noteworthy that this apparent constant is in the tens of micromolar range and higher than the submicromolar $Ca^{2+}$ affinity calculated for the regulatory EF-4 $Ca^{2+}$ binding site (*Table 4*). These differences may be due to different issues. First, the experimental conditions were different in each experiment: $Na^+$ concentration was 50 mM and 200 mM in the ITC and nucleotide exchange assays, respectively. Second, the nucleotide exchange assay requires 2 mM $MgCl_2$, and $Mg^{2+}$ competes with $Ca^{2+}$ for binding (*Burgoyne et al., 2019*; *Aravind et al., 2008*). Third, the apparent activation constant $K_{a-NCS-1}$ includes at least three different events: $Na^+$ dissociation from NCS-1 EF-4 (200 mM NaCl experimental conditions), dissociation of Ric-8A from the NCS-1/Ric-8A complex, and finally, binding of $Ca^{2+}$ to NCS-1. $Na^+$ dissociation from EF-4 and the disassembly of NCS-1 from Ric-8A are unfavorable contributions to NCS-1 $Ca^{2+}$ binding.

Finally, at this juncture, we have not explored the mechanism of $Ca^+$-induced intrinsic nucleotide exchange at $\Delta N31G\alpha_{i1}$.

## Discussion

Several studies have shown the structural determinants of Ric-8A chaperone and GEF activity for $G\alpha$ protein subunits (*Srivastava and Artemyev, 2019*; *McClelland et al., 2020*; *Seven et al., 2020*; *Zeng et al., 2019*; *Papasergi-Scott et al., 2023*). Ric-8A phosphorylation promotes $G\alpha$ recognition since it stabilizes a conformation that allows Ric-8A to trap the Ras-like domain of $G\alpha$ (*Figure 1B*). This facilitates the folding of the $G\alpha$ subunit and stabilizes a nucleotide-free state, which in turn would prepare the protein for GTP loading (*Srivastava and Artemyev, 2019*; *McClelland et al., 2020*; *Seven et al., 2020*). However, the mechanism by which the $G\alpha$ chaperone and GEF activity of Ric-8A could be downregulated by NCS-1 remained unclear. Here, we have combined biochemical, biophysical, and PPI assays with X-ray crystallography to shed light on the mechanism that keeps Ric-8A inactive.

The combination of the crystallographic work using Ric-8A peptides of different lengths, together with cell-based PPI assays with different Ric-8A and NCS-1 mutants, has enabled the discovery of the necessary and sufficient region of Ric-8A that is required for NCS-1 recognition, a region that is conserved between human and rat (*Figures 2 and 3*). This region corresponds to the two-helix bundle that constitutes the HEAT repeat 9 of the ARM-HEAT repeat domain, and a disordered region that, in the presence of $G\alpha$, forms an extended coil to permit phosphorylated S435 and T440 (in human S436 and T441) to attach to positively charged patches of Ric-8A (*Figures 1B and 7*; *Srivastava and Artemyev, 2019*; *McClelland et al., 2020*; *Seven et al., 2020*; *Zeng et al., 2019*). Binding of phosphorylated rRic-8A (pRic-8A) to $G\alpha$ does not substantially alter the conformation of the ARM-HEAT repeat domain, since the overall structure and orientation of repeats of uncomplexed pRic-8A and pRic-8A bound to $G\alpha$ are very similar, such that any major differences are confined to the polypeptide loop regions. As shown by the crystal structures presented here, the HEAT repeat 9 undergoes a substantial conformational change: helix a9 unfolds while helix b9 refolds, resulting in the named regions R1 and R2 (*Figure 3B* and *Figure 7B*). The relative orientation of R1 and R2 change with respect to a9 and b9. If helices b9 and R2 are superposed, helix a9 is rotated ~180 degrees with respect to R1 (*Figure 7B*). To illustrate the structural rearrangement, a morph movie has been produced starting at the $G\alpha$-bound and ending at the NCS-1-bound conformations (*Video 1*). The structural reorganization

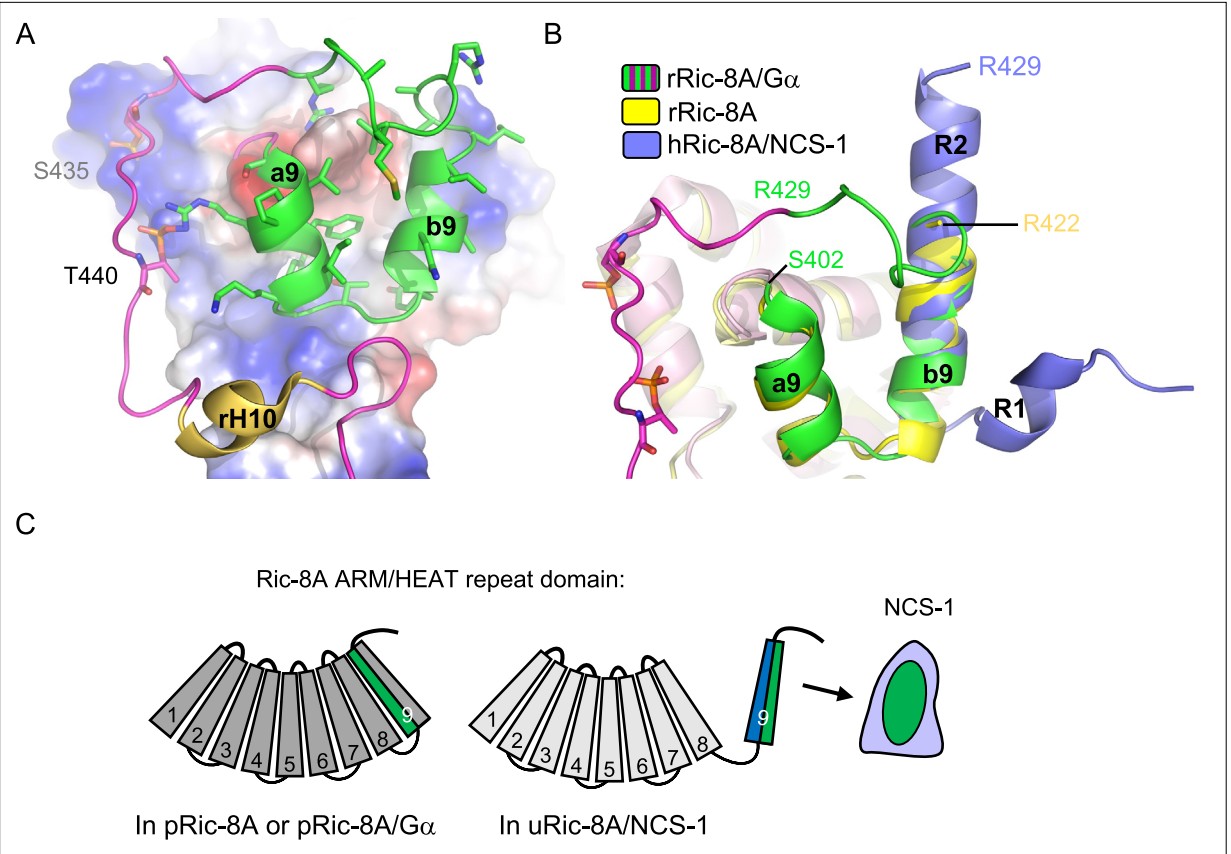

**Figure 7.** Structural reorganization of Ric-8A for NCS-1 recognition. (**A**) The structure of the rRic-8A/Gα$_{i1}$ complex (PDB: 6UKT, *McClelland et al., 2020*). Electrostatic potential surface representation of ARM-HEAT domain (repeats 1–8). The repeat 9 is shown as ribbons. The Ric-8A region present in the NCS-1/Ric-8A-P crystal structure is shown in green and side chains of the corresponding residues in stick mode. Phosphorylated S435 and T440 are indicated. (**B**) Superposition of the structures of Ric-8A bound to Gα (magenta and green, PDB: 6UKT, *McClelland et al., 2020*), uncomplexed rRic-8A (yellow, PDB: 6NMG, *Zeng et al., 2019*), Ric-8A peptide (light purple) bound to hNCS-1. Ric-8A helix R2 of the complex with NCS-1 was superposed with helix b9 of uncomplexed Ric-8A. (**C**) Schematic representation of Ric-8A ARM/HEAT repeat domain (repeats 1–9 are indicated) explaining the detachment of 9 for NCS-1/Ric-8A assembly. The redistribution of charged (blue) and hydrophobic residues (green) in Ric-8A repeat 9 generates the platform for NCS-1 recognition. Repeats 1–8 have been colored in different gray tonalities since previous studies have shown global changes within the ARM-HEAT repeat domain of unphosphorylated Ric-8A compared to the phosphorylated version (*Zeng et al., 2019*).

affects the distribution of hydrophobic and positively charged amino acids. In the Ric-8A/Gα complex, hydrophobic residues in the HEAT repeat 9 are facing the ARM repeat 8 to create contacts between the two repeats (*Figure 7A and C* and *Video 1*). In the NCS-1ΔH10/Ric-8A-P complex, the same hydrophobic residues are exposed to the solvent, affording their interaction with the hydrophobic crevice of NCS-1, while positively charged residues are exposed on the opposite face, generating an amphipathic structure (*Figure 3*, *Figure 3—figure supplement 2*, *Figure 7C* and *Video 1*). This would create electrostatic repulsions with repeat 8 and would force the detachment of repeat 9 from repeat 8, so that NCS-1 properly recognizes Ric-8A. The unfolding of helix a9 could act as a hinge to assist in the rearrangement that occurs when uRic-8A binds NCS-1.

The dramatic increase in thermal stability experienced by unphosphorylated Ric-8A upon binding to NCS-1 (>20°C in Ti; *Figure 2B*) is indicative of the strong interactions established in the complex between the two proteins. It also means that the free energy of binding would be high enough as to pay the energy penalty derived from the structural changes experienced by Ric-8A to give rise to the complex with NCS-1. Currently, it is unknown if the impact of NCS-1 binding to Ric-8A extends beyond the structural rearrangement that takes place at repeat 9 and its detachment from repeat 8. Nonetheless, it is worth noting that global changes within Ric-8A ARM-HEAT repeat domain have been observed when comparing the phosphorylated and unphosphorylated forms of Ric-8A-452 (*Zeng et al., 2019*). The combination of the crystallographic data with low-resolution

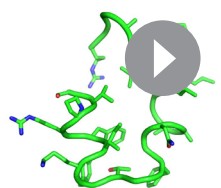

**Video 1.** Morph movie explaining the structural rearrangement of Ric-8A HEAT repeat 9 for NCS-1 recognition. Ric-8A residues 402–429 are shown starting at the Gα-bound and ending at the NCS-1-bound conformations. The view is the same as that in *Figure 7A*. Side chains are displayed in stick mode. While in the Ric-8A/Gα structure hydrophobic residues are at the back, facing the ARM-HEAT repeat domain (not shown), they rearrange and expose to the solvent to recognize NCS-1. The resulting structure is amphipathic and positively charged residues concentrate at the opposite side.
https://elifesciences.org/articles/86151/figures#video1

SAXS data suggested that there were quasi-rigid body angular displacements of three subdomains (constituted by repeats 1–4, 5–6, and 7–9) of the ARM-HEAT repeat domain (*Figure 7C*). Therefore, interactions between the phosphorylated residues and the C-terminal repeat 9 (*Figure 7A*) are translated into different contacts between the subdomains, along with a different global shape and curvature of the ARM-HEAT repeat domain (*Zeng et al., 2019*). The structure solution of the complete Ric-8A ARM-HEAT repeat domain bound to NCS-1 will allow to understand the extent of the rearrangement that Ric-8A suffers upon NCS-1 binding.

It is worth noting that most of the residues of Ric-8A implicated in the NCS-1/Ric-8A PPI are occluded in the absence of NCS-1 (*Figures 3 and 7*). The analysis of co-immunoprecipitation assays conducted with different NCS-1 and Ric-8A mutants point to the relevance of van der Waals interactions between the upper part of the NCS-1 crevice and the C-terminal half of Ric-8A helix R2 (*Figure 3D* and *Table 2*; see binding loss in NCS-1 W30A and Ric-8A L424A, M425A). In the complex with Gα, this region adopts an extended coiled structure and is attached to the ARM-HEAT repeat domain, due to the interactions with the phosphorylated Ric-8A residues (*Figure 7A*). In the absence of phosphorylation, this region would be exposed to the solvent and could acquire the helical structure needed to initiate the recognition of NCS-1. Initial contacts of the C-terminal part of helix R2 with NCS-1 could constitute the first steps in the recognition process and trigger the subsequent reorganization, which would imply the detachment of repeat 9 and complete the formation of helix R2 and, finally, the formation of helix R1 and the change in the relative orientation of R1 and R2 with respect to a9 and b9.

The dynamic C-terminal helix H10 of NCS-1 regulates the binding of Ric-8A by inserting into the hydrophobic crevice (*Figure 1A*; *Romero-Pozuelo et al., 2014*). Comparison of the structure of Ric-8A-P in complex with NCS-1ΔH10 with that of NCS-1 in which its helix H10 is inserted in the hydrophobic crevice explains why this helix constitutes a built-in competitive inhibitor of the NCS-1/Ric-8A interaction (*Figure 1—figure supplement 1*). The position of helix H10 completely overlaps with the Ric-8A region that is recognized by NCS-1 and displaces most of the H-bonds and contacts reported for the complex, including the C-terminus of helix R1, the N-terminus of helix R2, and the loop connecting them (*Figure 3—figure supplement 2B, C* and *Figure 1—figure supplement 1A*). Interestingly, the binding of Ric-8A does not cause a significant structural reorganization of NCS-1. Superimposition of the hNCS-1ΔH10/Ric-8A-P structure with that of free human NCS-1 (*Canal-Martín et al., 2019*) shows subtle changes: the NCS-1 helix H3 and the loop connecting the helices H3 and H4 rearrange to open up the cavity to accommodate helix R2 (*Figure 1—figure supplement 1C*). In addition, the loop connecting helices H7 and H8 helices, which interacts with the Ric-8A R1 region, undergoes a conformational change to permit the contacts between NCS-1 T135 and Ric-8A residues I407 and, to a lesser extent, V403 (*Figure 3—figure supplement 2A, D* and *Figure 1—figure supplement 1C*).

The data presented here show how $Ca^{2+}$ recognition and Ric-8A phosphorylation serve as determinants of NCS-1/Ric-8A recognition. Binding of $Ca^{2+}$ to the structural EF-hands EF-2 and EF-3 is necessary for protein recognition, while EF-4, the regulatory $Ca^{2+}$ binding site (*Aravind et al., 2008*), is free of $Ca^{2+}$, as shown in the crystal structure of the NCS-1ΔH10/Ric-8A-P complex (*Figure 4A*) and the in vitro reconstruction of the complex (*Figure 2A*). We show that a fully $Ca^{2+}$-loaded protein does not properly recognize Ric-8A (assembly assays, *Figure 2A*, and BLI, *Figure 4E* and *Table 3*), suggesting

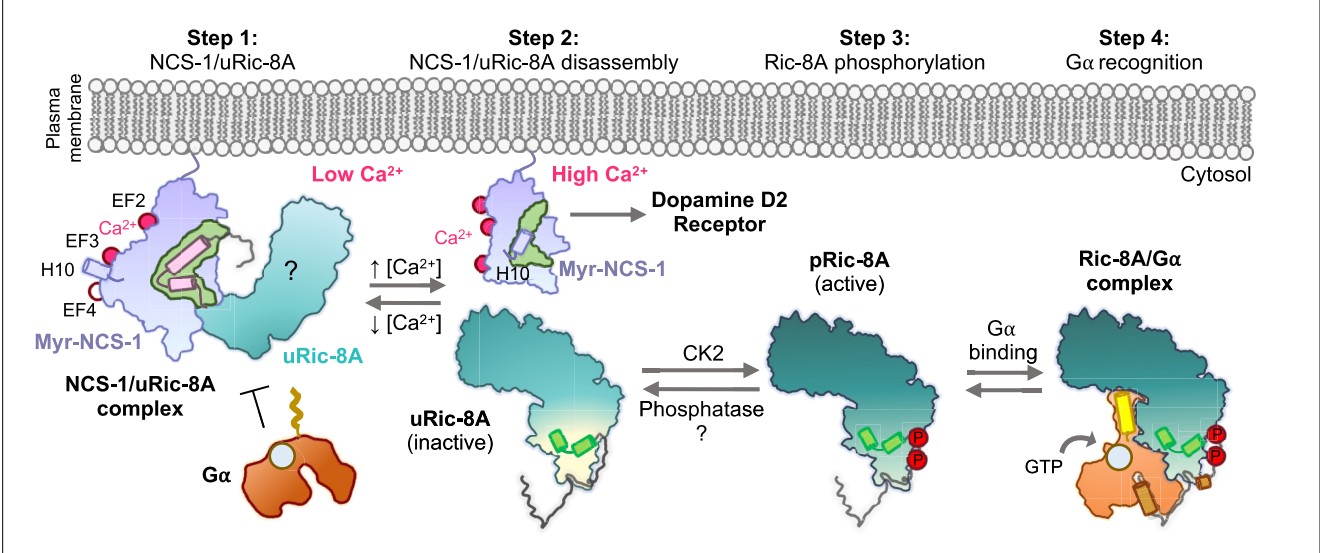

**Figure 8.** Schematic representation of the mechanism of Ric-8A activation regulated by NCS-1. Step 1: At low $Ca^{2+}$ concentrations NCS-1 interacts with unphosphorylated Ric-8A (uRic-8A), at the plasma membrane. NCS-1 protects Ric-8A from phosphorylation or $G\alpha$ subunit binding. Ric-8A ARM-HEAT repeat domain is labeled with a question mark since the structure in the context of the NCS-1 complex is unknown. Step 2: When $Ca^{2+}$ levels increase in the cytosol, NCS-1 binds $Ca^{2+}$ at EF-4 and the complex is disassembled. NCS-1 helix H10 inserts in the hydrophobic crevice (green) and would be ready for dopamine D2 receptor recognition. Inactive uRic-8A, free of NCS-1, repacks repeats 8 and 9 (helices a9 and b9 in light green) and S335 and T440 are phosphorylated (P) by casein kinase II (CK2) (Step 3). In this state, phosphorylated Ric-8A (pRic-8A) is now active, recognizes prefolded $G\alpha$ subunit and allows GTP loading (Step 4).

that in the cellular context, an increase in $Ca^{2+}$ concentration and the subsequent loading of EF-4 may trigger the disruption of the NCS-1/Ric-8A complex. This agrees with the previous cell-based PPI assays that showed a weakened affinity of NCS-1 for Ric-8A in $Ca^{2+}$ saturating conditions (*Romero-Pozuelo et al., 2014*). Moreover, our in vitro nucleotide exchange assays demonstrate a loss of Ric-8A GEF function when in complex with NCS-1, while nucleotide exchange rates are partially rescued following addition of $Ca^{2+}$ (*Figure 6*). With respect to the phosphorylation state of Ric-8A, NCS-1 binds to unphosphorylated Ric-8A protein (*Figures 2A and 5A*) and more importantly, in the context of the protein complex, NCS-1 protects Ric-8A from CK2-mediated phosphorylation (*Figure 5B*). In this situation, an increase of $Ca^{2+}$ levels would permit the disassembly of the NCS-1/Ric-8A complex, allowing release of Ric-8A from the membrane, and subsequent phosphorylation (*Figure 8*). This would promote Ric-8A activation for $G\alpha$ subunit binding and GEF activity. When inactivation of Ric-8A is required in the cellular context, Ric-8A dephosphorylation must precede NCS-1 binding, since the phosphorylated protein does not recognize NCS-1 properly (*Figures 5C and 8*). This can be explained as well in the context of the structure of phosphorylated Ric-8A bound to $G\alpha$, since the strong ionic interactions of phosphorylated S335 and T440 with the ARM-HEAT repeat domain would preclude the structural rearrangement of repeat 9 (*Figure 7A and C*). Cellular mechanisms of Ric-8A dephosphorylation are not understood. It has been proposed that in the cell, Ric-8A is constitutively phosphorylated (*Papasergi-Scott et al., 2018*). However, and in the light of the results presented here, in neurons and other tissues where NCS-1 is expressed (*Burgoyne et al., 2019*), Ric-8A phosphorylation may be under NCS-1 control and regulated by $Ca^{2+}$ signals. Since NCS-1 is a $Ca^{2+}$ sensor that is constantly bound to the membrane and close to $Ca^{2+}$ channels, changes of cytosolic $Ca^{2+}$ would promote the disassembly of the protein complex and the subsequent phosphorylation of Ric-8A by CK2 (*Figure 8*). Given that Ric-8A is a ubiquitously expressed protein, it is unknown if, in tissues where NCS-1 is not expressed, there are other $Ca^{2+}$ sensors in charge of the regulation of its activity. Furthermore, it is likely that NCS-1 interacts with Ric-8B, a chaperone for other $G\alpha$ (*Tall et al., 2003*; *Chan et al., 2011*), since the sequence and structure of the NCS-1 interacting region is conserved (*Papasergi-Scott et al., 2023*).

The transmission of the nerve impulse between neurons begins with the generation of an action potential in the axon initial segment. An action potential is a depolarization of the plasma membrane

that depends on Na$^+$ entry through specific voltage-gated channels and that propagates along the axon to the terminal. When the depolarization Na$^+$ wave reaches the axon terminal, voltage-dependent Ca$^{2+}$ channels open allowing the massive entrance of Ca$^{2+}$ which in turn triggers neurotransmitter release (*Llinás et al., 1981*; *Oheim et al., 2006*). Therefore, Na$^+$ precedes Ca$^{2+}$ waves. Here, we demonstrate that Na$^+$ decreases the affinity of NCS-1 for Ca$^{2+}$ (*Figure 4D* and *Table 4*). The crystal structure of the hNCS-1ΔH10/Ric-8A-P complex shows that NCS-1 binds Na$^+$ at EF-4, the regulatory Ca$^{2+}$ binding site. From a physiological point of view this finding is relevant since NCS-1/Ric-8A disassembly and Ric-8A activation would occur at a Ca$^{2+}$ signal that is higher than initially expected, ensuring the return to the Ric-8A inactive and NCS-1-complexed state when Ca$^{2+}$ levels decrease, due to sequestering by several high capacity, low specificity, Ca$^{2+}$ binding proteins.

This work also sheds light into the mechanism of specificity that allows NCS-1 to recognize different targets. NCS-1 interacts with several proteins related to G-protein signaling: Ric-8A and GPCRs such as the adenosine A2A, cannabinoid CB1, or the dopamine D2 receptors (*Kabbani et al., 2002*; *Navarro et al., 2012*; *Angelats et al., 2018*). The structure and Ca$^{2+}$ determinants of dopamine D2 receptor recognition by NCS-1 are known (*Figure 1*; *Pandalaneni et al., 2015*). It is relevant that the Ca$^{2+}$ signals that trigger protein-protein recognition are opposite: while the binding of NCS-1 to Ric-8A takes place at low Ca$^{2+}$ concentrations at which the functional site EF-4 is free of Ca$^{2+}$ (*Figures 3A and 4A*), binding to dopamine D2 receptor takes place at high Ca$^{2+}$ levels (*Figure 1A*). Therefore, while Ric-8A is being blocked by NCS-1, dopamine D2 receptor is free of NCS-1 and vice versa. Also, the role of the NCS-1 dynamic C-terminal helix H10 is different in the two molecular recognition processes, while NCS-1 uses this helical element to properly recognize dopamine D2 receptor (*Pandalaneni et al., 2015*), the helix H10 negatively regulates the binding to Ric-8A. Finally, if the crystal structure of the NCS-1ΔH10/Ric-8A-P complex is compared with other NCS proteins bound to their corresponding targets, the complex that is structurally most similar is the KChIP1/Kv4.3 pair, with an elongated helix resembling the R2 bound to Ric-8A (*Figure 1—figure supplement 1D*). Interestingly, the N- to C-orientation of the helix is opposite, and no extra helical segment (R1) is recognized. In addition, the Ca$^{2+}$ content is different (*Figure 1A*; *Pioletti et al., 2006*). The number of structures that have been determined of NCS proteins bound to different targets continues to grow. Comparison of these structures show that, while all of the ligands use the same NCS hydrophobic crevice, differences in Ca$^{2+}$ site ligation, role of the helix H10, and the loop that connects EF-3 and EF-4 in NCS-1 shape the crevice to recognize, very specifically, different target protein motifs. In this sense polar interactions between NCS-1 and the target protein play also an important role to ensure specificity and affinity (*Figure 1A* and *Figure 1—figure supplement 1*).

Small-molecule PPI modulators with therapeutic potential have been discovered in the past few years (*Mansilla et al., 2017*; *Canal-Martín et al., 2019*; *Roca et al., 2018*). They are able to inhibit (e.g. the phenothiazine FD-44) or stabilize (e.g. the acylhydrazone 3b) the formation of the NCS-1/Ric-8A complex and in doing so, they regulate synapse number and function and show a promising prospect in the treatment of neurodevelopmental disorders (*Mansilla et al., 2017*; *Cogram et al., 2022*) and neurodegenerative diseases (*Canal-Martín et al., 2019*). It has been suggested that these small molecules, which bind to the same NCS-1 site despite their opposite activity, are not Ric-8A competitors. Instead, they have been suggested to be allosteric modulators that contribute to the stabilization of the dynamic helix H10 inside (inhibitors) or outside (stabilizers) the crevice, favoring NCS-1 conformations that hinder or allow the entrance of Ric-8A and the subsequent formation of the protein complex. Supporting this, the modulator FD-44 does not inhibit the formation of the NCS-1/Ric-8A complex when the dynamic helix H10 is deleted. The structure of the hNCS-1ΔH10/Ric-8A-P complex explains the molecular mechanism of action of the regulatory compounds. Comparison of the different complexes shows that the binding sites for these molecules overlap with the C-terminal half of helix R2 (*Figure 1—figure supplement 1A, B*). With respect to the inhibitors, hydrophobic interactions of FD-44 with the helix H10 stabilize the crevice-inserted helix H10 conformation of NCS-1. The combination of the compound plus helix H10 makes the Ric-8A interacting region of NCS-1 completely unavailable (*Figure 1—figure supplement 1A*). On the contrary, for the PPI stabilizer, the polar characteristics of the molecule hinders the approach of the helix H10 to the crevice, making the C-terminal part of the cavity, which is more relevant in terms of the PPI, available for Ric-8A recognition (*Figure 1—figure supplement 1B*). The structure of hNCS-1ΔH10/Ric-8A-P does not support the existence of a ternary NCS-1/Ric-8A/stabilizer complex, since there would be no room

for the repositioning of the compound. Therefore, the binding of Ric-8A to NCS-1 would displace modulator 3b from the crevice, which would be likely given the moderate affinity of the compound (*Canal-Martín et al., 2019*). Finally, we believe that the high-resolution data on the NCS-1/Ric-8A PPI interface presented here will be essential to rationally develop improved compounds with better PPI regulatory properties and selectivity, since the relevant interactions between NCS-1 and Ric-8A have been finally revealed. The fact that the NCS-1 $Ca^{2+}$ and structural determinants are different in recognizing Ric-8A and the dopamine D2 receptor is conceptually relevant, since opens the path to the design of selective drugs that target specifically these neuronal pathways.

## Materials and methods
### Cloning, expression, and purification of proteins

*NCS-1 constructs:* Human full-length NCS-1 (100% protein sequence identity with the rat variant) was cloned in the pETDuet vector (*Canal-Martín et al., 2019*). A stop codon was introduced after residue P177 to generate the NCS-1ΔH10 construct and using the IVA cloning strategy (*García-Nafría et al., 2016*; *Watson and García-Nafría, 2019*). The proteins were overexpressed in *Escherichia coli* (BL-21*) as reported (*Canal-Martín et al., 2019*; *Baños-Mateos et al., 2014*). Briefly, cells were resuspended in lysis buffer (50 mM HEPES pH 7.4, 100 mM KCl, 1 mM 1,4-dithiothreitol (DTT), 0.1 mM phenylmethylsulfonyl fluoride (PMSF), 10 µg/ml DNAse) and disrupted by sonication. The lysate was centrifuged at 16,000 rpm in a SS-34 rotor (4°C, 45 min). One mM $CaCl_2$ was added to the clarified supernatant and the resulting solution was injected into a Hi Trap Phenyl FF hydrophobic column (Cytiva) preequilibrated with lysis buffer plus 1 mM $CaCl_2$. The column was washed with 5 column volumes of HIC-A buffer (20 mM Tris pH 7.9, 1 mM $CaCl_2$, 1 mM DTT). Protein elution was achieved applying a EGTA gradient with HIC-B buffer (20 mM Tris pH 7.9, 2 mM EGTA, 2 mM DTT). NCS-1ΔH10 elution occurred at 1.2 mM EGTA, while full-length NCS-1 elution occurred at 1.4 mM EGTA. Protein quality was evaluated by nano-DSF (see below) and SDS-PAGE. Fully $Ca^{2+}$-loaded protein was prepared by dialyzing the EGTA-eluted sample against HIC-A buffer. Then, the sample was loaded in an anion exchange HP Q column (Cytiva) and eluted with a NaCl gradient using QB buffer (20 mM Tris pH 7.9, 1 mM $CaCl_2$, 500 mM NaCl, 1 mM DTT). NCS-1 elution occurred at 175 mM NaCl.

N-terminally hexahistidine-tagged full-length human NCS-1 (His-NCS-1) was expressed in a pET28a+ vector using BL21(DE3) pLysS *E. coli*. Overnight cultures were grown in 100 ml LB media containing 100 µg/ml kanamycin at 200 rpm and 37°C. After ~16 hr cells were pelleted at 2200 × *g* for 10 min at 4°C using a benchtop Sorvall Legend RT. Resuspended pellets were added to 1 l 2xYT media containing 100 µg/ml kanamycin and incubated at 37°C and 200 rpm until an $OD_{600}$ of 0.8–1 was achieved and cells were induced with 0.3 mM isopropyl β-D-1-thiogalactopyranoside at 16°C. Approximately 18 hr post-induction, cells were pelleted at 12,000 × *g* for 15 min in a Sorvall RC6+ Centrifuge and cell pellets were stored at –80°C. Pellets were resuspended in 50 ml (per l of cell pellet) of 50 mM HEPES pH 7.4, 100 mM KCl, 2 mM beta-mercaptoethanol, 10 µg/ml DNase, and lysis was performed with an Avestin Emulsiflex-C5 homogenizer. Lysate was cleared for 40 min at 4°C and 35,000 rpm with a Beckman Coulter Optima XE-90 ultracentrifuge and a Type 45 TI rotor and the supernatant was loaded on a HisTrap FF crude column. NCS-1 eluted at 25% during a 0–100% gradient to 50 mM Tris pH 8, 300 mM imidazole, 1 mM $CaCl_2$, 2 mM beta-mercaptoethanol. Eluted His-NCS-1 was dialyzed overnight in HIC-A buffer and dialyzed His-NCS-1 was loaded on a 5 ml HiTrap Phenyl HP column and eluted, as described above, in HIC-B buffer.

*Ric-8A constructs:* Rat Ric-8A-452 (residues 1–452) was previously cloned in pET28a vector (*Kant et al., 2016*). Introduction of stop codons after residues G423 and G432 resulted in the so-called Ric-8A-423 and Ric-8A-432 constructs. Proteins were expressed in Rosetta2 pLysS cells and purified similarly to previously reported methods (*Zeng et al., 2019*). Briefly, cells were resuspended in lysis buffer (50 mM Tris pH 8, 250 mM NaCl, 20 mM imidazole, 5% glycerol, 2 mM beta-mercaptoethanol) with 0.1 mM PMSF, 10 µg/ml DNAse and protease inhibitors (cOmplete, EDTA-free cocktail, Roche), and disrupted with a sonicator. After centrifugation (45 Ti rotor at 30,000 rpm and 4°C for 40 min), the clarified supernatant was loaded in a Nickel-affinity column (HisTrap FF, Cytiva). The column was washed with 10 volumes of lysis buffer and protein elution was achieved with a gradient with NiB buffer (50 mM Tris pH 8, 250 mM NaCl, 500 mM imidazole, 5% glycerol, 2 mM beta-mercaptoethanol), after an extra wash step with 7% NiB buffer. Protein sample was next dialyzed in GF buffer (50 mM Tris

pH 8, 150 mM NaCl, 5% glycerol, 2 mM beta-mercaptoethanol) for imidazole removal. Tobacco etch virus (TEV) protease was used to cleave the hexahistidine-tag off using a 1:40 molar ratio (TEV:His-Ric-8A) during 16 hr. After TEV treatment, the sample was incubated with $Ni^{2+}$-chelated Sepharose HP beads (Cytiva) to get rid of the uncleaved protein. A final polishing step was performed in a size exclusion column (HiLoad 16/600 Superdex 200 pg, Cytiva) preequilibrated in GF buffer. Protein quality was evaluated by nano-DSF (see below) and SDS-PAGE.

N-terminally hexahistidine-tagged rat Ric-8A residues 1–491, Ric-8A-491, was expressed and purified as previously described in a pET28a vector with BL21(DE3) pLysS *E. coli* (*Zeng et al., 2019*; *Kant et al., 2016*; *Thomas et al., 2011*). Post hexahistidine-tag removal by TEV protease and purification by Source 15Q column (*McClelland et al., 2020*), Ric-8A-491 was buffer exchanged into 50 mM Tris pH 8, 250 mM NaCl, and 1 mM Tris(2-carboxyethyl) phosphine (TCEP) for assembly of protein complex.

*Gα construct:* N-terminally glutathione-*S*-transferase tagged rat $G\alpha_{i1}$ with a 31 residue N-terminal truncation ($\Delta N31G\alpha_{i1}$) was expressed from a pDest15 vector in BL21(DE3) RIPL *E. coli* and purified as previously described (*Thomas et al., 2011*; *Thomas et al., 2008*).

## Assembly of protein complexes

### NCS-1ΔH10/rRic-8A-452 complexes

#### $Ca^{2+}$-free conditions (Assembly (i))

Pure rat His-Ric-8A-452 (50 mM Tris pH 8, 250 mM NaCl, 5% glycerol, 2 mM beta-mercaptoethanol) was mixed with NCS-1ΔH10 (20 mM Tris pH 7.9, 1.2 mM EGTA, 1 mM DTT) in a 1:1.9 (Ric-8A:NCS-1) molar ratio. Final NaCl and EGTA concentrations were adjusted to 125 mM and 0.6 mM, respectively. To ensure absence of $Ca^{2+}$, the mixture was dialyzed against buffer 50 mM Tris pH 8, 200 mM NaCl, 2 mM EGTA, 1 mM TCEP, with and without 1 mM $MgCl_2$ (2 changes, first after 4 hr and second for 16 hr). The final sample was concentrated and subjected to a SEC in the same buffer and using a Superdex 200 HR 10/300 column (Cytiva). 12% SDS-PAGE gels were run to identify the composition of eluted samples.

#### High $Ca^{2+}$ conditions (Assembly (ii))

Protein mixture was performed as above but the purified fully $Ca^{2+}$-loaded NCS-1ΔH10 (20 mM Tris pH 7.9, 2 mM $CaCl_2$, 1 mM DTT) was used instead. The final NaCl and $CaCl_2$ concentrations in the protein mixture were 125 mM and 2 mM, respectively. After 1 hr incubation, the sample was concentrated and subjected to a SEC. as described above, but the column was equilibrated in buffer 50 mM Tris pH 8, 200 mM NaCl, 2 mM $CaCl_2$, 1 mM TCEP.

#### Dialysis from EGTA to $Ca^{2+}$ conditions (Assembly (iii))

Protein mixture was the same as that for $Ca^{2+}$-free conditions and the resulting sample (at 0.6 mM EGTA) was dialyzed against buffer containing 50 mM Tris pH 8, 200 mM NaCl, 2 mM $CaCl_2$, 1 mM TCEP. This assembly was performed with both the un-phosphorylated and phosphorylated variants of His-Ric-8A-452. The same assembly was also performed in a $K^+$ containing buffer. For this, unphosphorylated His-Ric-8A-452 was first dialyzed against buffer containing 50 mM Tris pH 8, 250 mM KCl, 5% glycerol, 2 mM beta-mercaptoethanol, to replace KCl for NaCl. Protein mixture, dialysis, and gel filtration were carried out substituting KCl for NaCl.

### His-NCS-1/rRic-8A-491 complex

Complexes used for guanine nucleotide exchange assays were prepared as described in Assembly (iii) and purified by SEC on a HiLoad 16/600 Superdex 200 pg in 50 mM HEPES pH 8, 200 mM NaCl, 1 mM TCEP. NCS-1/rRic-8A-491 complexes prepared for assays conducted at less than or equal to 1 μM $Ca^{2+}$ were formulated, purified, and assayed using buffers made with HPLC-grade water (RPI).

## Assembly, crystallization, diffraction data collection, and structure solution of hNCS-1ΔH10/Ric-8A-P complexes

Three highly pure (>95%) HPLC-verified Ric-8A peptides were purchased from GenicBio for structural studies. They ranged from residue 400 to residues 423 (P1), 429 (P2), and 432 (P3) (*Figure 1B*). Lyophilized peptides were solubilized in HIC-B buffer and mixed with purified NCS-1ΔH10 in a 1:10

(protein:peptide) molar ratio (final EGTA concentration 1.7 mM). The mixture was dialyzed against a buffer containing 20 mM Tris pH 8, 0.5 mM $CaCl_2$, 0.5 mM DTT (2 changes, first after 4 hr, second for 16 hr). Thermal stability of the final samples was evaluated by nano-DSF. The final sample was concentrated to 20 mg/ml with a Vivaspin 2 device (2 kDa cutoff, Sartorius).

Crystallization screenings were set with an Oryx8 robot (Douglas Instruments) at 4°C, using the sitting drop vapor diffusion method and mixing equal volumes of protein complex and precipitant. Initial crystals were obtained with P2 and P3 peptides and solution from JBScreen Classic (Jena Bioscience) and INDEX (Hampton Research) crystallization screenings. Diffracting crystals obtained with P2 peptide grew using microseeding techniques and precipitant solution 25% PEG 4000, 100 mM NaAc pH 5, and 100 mM $MgCl_2$. Crystals with peptide P3 grew in two different conditions: 30% PEG 4000, 100 mM NaAc pH 4.6, 100 mM $MgCl_2$ and 25% PEG 3350, 100 mM NaAc pH 4.5. Crystals were cryoprotected adding 30% (vol/vol) glycerol to the precipitant solutions and flash-frozen in $N_2$(l).

Diffraction data were collected at 100 K and 0.979 Å wavelength at ALBA synchrotron radiation source (BL13 beamline) (*Table 1*). Data were processed with AutoPROC using the extended anisotropic method (*Vonrhein et al., 2011*). The first structure was solved by molecular replacement with Phaser (*McCoy et al., 2007*), using with data from P2 peptide crystals (Structure 1). As search model, the structure of hNCS-1 (PDB: 6QI4), lacking the C-terminal helix H10, was used (*Canal-Martín et al., 2019*). Successive cycles of automatic refinement with Phenix (*Adams et al., 2010*) and manual building with Coot (*Emsley and Cowtan, 2004*) were performed. The refined structure was used to solve Structures 2 and 3, using Fourier differences calculations. The final models were validated with Molprobity (*Williams et al., 2018*). Details on data processing and refinement are shown in *Table 1*. The structures were analyzed using different programs from the CCP4 package (*Winn et al., 2011*) and the PISA server (*Krissinel and Henrick, 2007*). Images were prepared with PyMOL (*Schrödinger, 2015*). The final structures were deposited in the PDB with codes: Structure 1 (8ALH), Structure 2 (8AHY), Structure 3 (8ALM).

## Thermal shift assay

Label-free thermal shift assays with hNCS-1 full-length, hNCS-1ΔH10, rRic-8A-452, NCS-1ΔH10/rRic-8A-452, NCS-1ΔH10/Ric-8A-P2 peptide, and NCS-1ΔH10/Ric-8A-P3 peptide were performed using a Tycho NT.6 instrument (NanoTemper Technologies). This nano-DSF instrument records the protein's intrinsic fluorescence at 330 nm and 350 nm while heating the sample from 35°C to 95°C at a rate of 30°C/min. Tycho NT.6 automatically generates thermal unfolding profiles by representing the fluorescence ratio (350/330 nm) as a function of increasing temperature, giving insights on the thermal stability of the protein and allowing the analysis of interactions effects on relative stability. Tycho NT.6 software detects and identifies the inflection temperature (Ti) of the unfolding transition/s. A peak in the first derivative view corresponds to the detected Ti of the test sample.

Proteins at 10 µM in their corresponding final buffers (see above) were measured using NanoTemper capillaries. Three independent replicates were performed for each sample. The mean Ti and standard error of the mean (SEM) values were calculated for each sample.

## Phosphorylation assays

Purified rRic-8A-452 and NCS-1ΔH10/rRic-8A-452 were phosphorylated with CK2 (New England Biolabs) as previously described by *McClelland et al., 2020*. Briefly, 2 mg of each were dialyzed (2 changes, 2 hr and o/n, 4°C) in prephosphorylation buffer (50 mM Tris pH 8, 150 mM NaCl, 2 mM $CaCl_2$, 1 mM TCEP). Samples were mixed 1:1 in 2× reaction buffer (100 mM Tris pH 8, 200 mM NaCl, 20 mM $MgCl_2$, 2 mM EGTA, 1 mM DTT). Half of the samples was subjected to phosphorylation by adding 300 U CK2 and 5 mM ATP. Reactions were allowed to proceed for 16 hr and 18°C. The other half of samples were treated similarly but CK2 and ATP were not added (non-phosphorylated sample; controls).

To distinguish between phosphorylated and non-phosphorylated proteins, anion exchange chromatography was performed with the non-phosphorylated samples (controls) and those subjected to CK2 treatment. Samples were dialyzed in RV buffer (50 mM Tris pH 8, 125 mM NaCl, 1 mM $CaCl_2$, 1 mM DTT) and injected into an anion exchange HiTrap Q HP column (Cytiva) preequilibrated with QA buffer (50 mM Tris-HCl pH 8, 75 mM NaCl, 1 mM $CaCl_2$, 1 mM DTT). Protein elution was achieved with a gradient using QB buffer (50 mM Tris-HCl pH 8, 500 mM NaCl, 1 mM $CaCl_2$, 1 mM DTT). Fractions

from each peak were collected and analyzed by SDS-PAGE. Presence of phosphorylation in rRic-8A-452 and NCS-1ΔH10/rRic-8A-452 complex was additionally verified in a phosphoprotein assay by LC-MS/MS.

## Phosphoprotein analysis by LC-MS/MS

The phosphoprotein assay was divided into three different steps: (1) in-gel sample digestion; (2) phosphopeptide purification; and (3) protein identification by tandem mass spectrometry.

*In-gel sample digestion:* Gel band samples from 1D gel separation and Coomassie staining, were automatically in-gel digested. Gel bands were excised, cut into cubes (1 mm²), deposited in 96-well plates, and automatically processed in an OT-2 digestor (Opentrons, NY, USA). The digestion protocol used was based on *Shevchenko et al., 1996* (*Pineiro et al., 2019*) with minor variations: gel plugs were washed first with 50 mM ammonium bicarbonate and second with acetonitrile, prior to reduction and alkylation (5 mM tris(2-carboxyethyl)phosphine) and 10 mM chloroacetamide in 50 mM ammonium bicarbonate solution, at 56°C for 30 min. Gel pieces were then rinsed first with 50 mM ammonium bicarbonate, and second with acetonitrile, and then were dried under a stream of nitrogen. Pierce MS-grade trypsin (Thermo Fisher Scientific, MA, USA) was added at a final concentration of 16 ng/µl in 50 mM ammonium bicarbonate solution, and the digestion took place at 37°C for 2 hr. Peptides were recovered in 50% ACN/0.5% FA, dried in speed-Vac and kept at –20°C until phosphopeptide enrichment.

*Phosphopeptide purification:* Phosphopeptide enrichment procedure utilized two concatenated in-house packed microcolumns, immobilized metal affinity chromatography, and Oligo R3 polymeric reversed-phase that provided selective purification and sample desalting prior to LC-MS/MS analysis, and was performed as previously reported (*Navajas et al., 2011*).

*Protein identification by tandem mass spectrometry (LC–MS/MS Exploris 240):* The peptide samples were analyzed on a nano-liquid chromatography system (Ultimate 3000 nano HPLC system, Thermo Fisher Scientific) coupled to an Orbitrap Exploris 240 mass spectrometer (Thermo Fisher Scientific). Samples (5 µl) were injected on a C18 PepMap trap column (5 µm, 100 µm ID × 2 cm, Thermo Scientific) at 20 µl/min, in 0.1% formic acid in water, and the trap column was switched online to a C18 PepMap Easyspray analytical column (2 µm, 100 Å, 75 µm ID × 50 cm, Thermo Scientific). Equilibration was done in mobile phase A (0.1% formic acid in water), and peptide elution was achieved in a 30 min gradient from 4% to 50% B (0.1% formic acid in 80% acetonitrile) at 250 nl/min. Data acquisition was performed using a data-dependent top 15 method, in full scan positive mode (range of 350–1200 m/z). Survey scans were acquired at a resolution of 60,000 at m/z 200, with normalized automatic gain control (AGC) target of 300% and a maximum injection time (IT) of 45 ms. The top 15 most intense ions from each MS1 scan were selected and fragmented by higher-energy collisional dissociation (HCD) of 28. Resolution for HCD spectra was set to 15,000 at m/z 200, with AGC target of 75% and maximum ion IT of 80 ms. Precursor ions with single, unassigned, or six and higher charge states from fragmentation selection were excluded.

MS and MS/MS raw data were translated to mascot general file (mgf) format using Proteome Discoverer (PD) version 2.5 (Thermo Fisher Scientific), and searched using an in-house Mascot Server v. 2.7 (Matrix Science, London, UK) against an in-house database including Ric-8A protein sequence along with common laboratory protein contaminants. Search parameters considered fixed carbamidomethyl modification of cysteine, and the following variable modifications: methionine oxidation, phosphorylation of serine/threonine/tyrosine, and deamidation of asparagine/glutamine. Peptide mass tolerance was set to 10 ppm and 0.02 Da, in MS and MS/MS mode, respectively, and three missed cleavages were allowed. The Mascot confidence interval for protein identification was set to ≥95% (p<0.05) and only peptides with a significant individual ion score of at least 30 were considered.

## Binding of NCS-1 to Na⁺, K⁺, and Ca²⁺

*Intrinsic fluorescence titration assay:* Because NCS-1 contains two tryptophan residues, W30 and W103, the protein shows intrinsic fluorescence when excited at 295 nm. These residues are located at EF-1 (W30) and EF-3 (W103). The latter is located at helix H6, which is in contact with helix H9, part of the EF-4 motif, where Na⁺ binds (*Figure 3—figure supplement 2*). Tryptophan emission fluorescence is very sensitive to the 3D environment and even subtle structural rearrangements have an effect on both the emission intensity and the spectra's maximum wavelength. This technique has been used

previously to study the binding of different ligands to NCS-1 by monitoring changes in the fluorescence emission intensity at increasing amounts of the compound under study (*Mansilla et al., 2017*; *Canal-Martín et al., 2019*; *Roca et al., 2018*). Tryptophan emission fluorescence of EGTA-purified full-length NCS-1 was recorded at 10 µM in buffer containing 20 mM Tris pH 8, 100 µM EGTA, 1 mM DTT, and 0–300 mM NaCl or 0–300 mM KCl. Data were acquired with a Tycho NT.6 equipment (NanoTemper Technologies). The emission fluorescence intensity was recorded at 330 nm and 35°C. Fluorescence intensities were normalized as $(I_0 - I)/I_0$. Three independent experiments were performed. Mean ± SEM values were represented at different $Na^+$ and $K^+$ concentrations. The apparent dissociation constant was calculated by using a least squares algorithm to fit the experimental data to a 1:1 stoichiometry model (*Mansilla et al., 2017*). The fitting was performed with KaleidaGraph Data Analysis Program (*Tellinghuisen, 2000*).

*ITC*: $Ca^{2+}$ binding was characterized by ITC at 25°C using a VP-ITC microcalorimeter (GE Healthcare, Northampton, MA, USA) with a cell volume of 1.4619 ml in $Na^+$ or $K^+$ containing buffers (20 mM Tris pH 7.9, 2 mM EGTA, 150 mM NaCl, or 150 mM KCl). Before measurements, EGTA-purified full-length hNCS-1 (25 µM) was dialyzed in parallel against the above buffers (3×300 ml; 2 hr, 2 hr, and 20 hr) and then against the same buffers without EGTA. Protein solutions at 110 µM were loaded into the calorimetric cell and titrated by stepwise injections of a 1.5 mM $CaCl_2$ solution prepared in the final dialysate. Dilution heats, evaluated separately, were found to be negligible. The binding isotherms were fit by nonlinear regression analysis using the AFFINImeter software (*Pineiro et al., 2019*) using the model builder to create a sequential binding model with three different binding sites:

$$M+L \; \underset{K_1}{\overset{+L}{\rightleftarrows}} \; ML_1 \; \underset{K_2}{\overset{+L}{\rightleftarrows}} \; ML_2 \; \underset{K_3}{\overset{+L}{\rightleftarrows}} \; ML_3$$

where M and L refer to NCS-1 and $Ca^{2+}$, respectively. Values of $K_i$ and $\Delta H_i$, the stoichiometric dissociation constant and the enthalpy change for step i (i=1–3), were directly determined from data fitting. The free energy change of binding was calculated as $\Delta G_i = -RT \ln(1/K_i)$ (R=1.986 cal/mol/K).

## Biolayer interferometry

The $Ca^{2+}$ dependence of NCS-1 binding to Ric-8A was assessed by BLI in a single-channel BLItz system (ForteBio). This optical label-free technique allows the measurement of macromolecular interactions by analyzing interference patterns of white light reflected from the surface of a biosensor tip, where one of the molecules is immobilized. The high concentration of immobilized molecules at the tip allows the detection of low-affinity binders and therefore the study of weak interactions. Changes in the number of molecules interacting with the immobilized molecules bound to the biosensor tip cause a shift in the interference pattern ($\Delta \lambda$) that is measured in real time. An apparent equilibrium constant, $K_d$, can be calculated from the dissociation and association rate constants obtained from the experimental profiles (*Sultana and Lee, 2015*).

N-terminally His-tagged NCS-1 was immobilized in Ni-NTA biosensors (Sartorius) and binding to Ric-8A-P3 peptide was tested at increasing concentrations of $Ca^{2+}$. To avoid $Ca^{2+}$ loading of the functional $Ca^{2+}$ binding site, buffers were prepared with decalcified milli-Q water to avoid any $Ca^{2+}$ traces (LiChrosolv, Merck). EGTA-purified His-NCS-1 (1.9 mM EGTA) was diluted to 110 µM EGTA with $Ca^{2+}$-free buffer (50 mM Tris pH 7.9, 125 mM NaCl). Next, the protein was concentrated to 2 mg/ml and dialyzed against $Ca^{2+}$-free buffer to remove EGTA. To verify that the functional EF-4 $Ca^{2+}$ binding site was empty, a nano-DSF assay of the final His-NCS-1 protein was performed (*Figure 4—figure supplement 2*), showing a $T_i$ similar to that of NCS-1ΔH10 in EGTA (*Figure 2B*). Ric-8A-P3 peptide was solubilized in $Ca^{2+}$-free buffer. The final concentration of protein and peptide were set to 5 µM and 30–150 µM. $CaCl_2$ was added to protein, peptide, and buffer to achieve final $Ca^{2+}$ concentrations of 250, 375, and 425 nM to study the $Ca^{2+}$ dependence of the protein-peptide interaction. NCS-1 immobilization sequence was as follows: (1) baseline (buffer, 30 s), (2) loading (His-NCS-1 in buffer, 300 s), and (3) equilibration (buffer, 300 s) (*Figure 4—figure supplement 2*). A control experiment was performed previously to verify that Ric-8A-P3 peptide does not bind to the Ni-NTA biosensor in the absence of NCS-1 (*Figure 4—figure supplement 2*). Also, the interaction of NCS-1 and Ric-8A-P3 was tested at different peptide concentrations to verify the specific binding of the peptide to the protein (*Figure 4—figure supplement 2*). For the NCS-1/Ric-8A interaction assay, and once His-NCS-1 was

bound to the tip, the sequence was: (1') baseline (buffer, 30 s), (2') association (Ric-8A-P3 in buffer, 300 s), and (3') dissociation (buffer, 300 s). A concentration of 50 μM Ric-8A-P3 was selected to study the protein-peptide interaction at increasing concentrations of $Ca^{2+}$. Three independent experiments were performed for each $Ca^{2+}$ concentration. Sensograms were analyzed and fit with the BLItz Pro software and apparent dissociation constant $K_d$ was calculated from fitted data and considering 1:1 equilibrium with the same software. Apparent $K_d$ values are represented as mean ± SEM in *Table 3*.

## Co-immunoprecipitations

Human NCS-1 and V5-tagged Ric-8A construct were previously described (*Mansilla et al., 2017*). Using the IVA cloning strategy (*García-Nafría et al., 2016*; *Watson and García-Nafría, 2019*) deletion constructs were prepared ending at residues G424 (hRic-8A-424, which corresponds to G423 in the rat variant) and G433 (hRic-8A-433, in rat, G432). Furthermore, a full-length hRic-8A mutant (S436A and T441A) was prepared to avoid phosphorylation of the protein at these sites. To verify the NCS-1/Ric-8A PPI interface, several hNCS-1 ((1) D37A, Y52A; (2) R148A, R151A; (3) D37A, R148A, R151A; (4) D37A, Y52A, R148A, R151A; and (5) W30A) and hRic-8A ((1) T411A, Y413A, N415A and (2) L425A, M426A) mutant proteins were generated. The numbering of the rat Ric-8A sequence has been maintained in *Figure 3D* for proper structural comparison. Constructs were cotransfected into HEK293 cells using Lipofectamin 2000 (Thermo) following the manufacturer's instructions. HEK293T cells were purchased from ATCC and authentication was provided by manufacturer. HEK293T cells were negative in mycoplasma contamination, as tested regularly. 48 hr after transfection cells were lysed in lysis buffer (150 mM NaCl, 1.0% Nonidet P-40, 50 mM Tris pH 8.0). Lysates were then incubated overnight (12 hr) at 4°C with mouse anti-NCS-1 (1:500; Cell Signaling). Samples were subsequently incubated for 2 hr with Protein-G-Sepharose (Sigma-Aldrich). After three washes with lysis buffer, proteins were eluted from the Sepharose and analyzed by western blot following standard procedures; 10% of the lysate before immunoprecipitation was run as input. Mouse anti-V5 (1:5000; Thermo) and rabbit anti-NCS-1 (1:2000; Cell Signaling) antibodies were used for western blot. The immunoprecipitation blot was incubated with anti-mouse TrueBlot (Rockland) as secondary antibody to avoid heavy-/light-chain antibody interference. Input blots were incubated with anti-mouse or anti-rabbit HRP-conjugated antibodies (Sigma 1:5000). HRP activity from the secondary antibodies was revealed with ECL (Promega) and pictures were taken with Chemidoc (Bio-Rad). Bands densitometry was performed using ImageJ (*Schneider et al., 2012*). A paired Student's t-test was used to compare mean values of three independent experiments. Graph and statistical analysis were performed using GraphPad Prism (GraphPad Software, Inc, USA).

## Guanine nucleotide exchange assays

Nucleotide exchange assays were carried out at 20°C using a LS55 luminescence spectrometer (Perkin Elmer) with 5 nm slit widths (Ex/Em 295 nm/345 nm). Assays were conducted by measuring the change in rat $\Delta N31G\alpha_{i1}$ tryptophan fluorescence in the presence or absence of rRic-8A-491, His-NCS-1/rRic-8A-491, or His-NCS-1 as previously described (*McClelland et al., 2020*; *Kant et al., 2016*). All assays were conducted in 50 mM HEPES pH 8, 200 mM NaCl, 2 mM $MgCl_2$, 1 mM TCEP. His-NCS-1/rRic-8A-491 was preincubated with $r\Delta N31G\alpha_{i1}$ and 0, 0.25, 0.375, 0.425, 1, 10, 25, 50, and 500 μM $CaCl_2$ in a quartz fluorescent cuvette prior to addition of GTPγS (guanosine 5'-*O*-[gamma-thio]-triphosphate). Assays with rRic-8A-491 and His-NCS-1 were also performed at 0, 1, 10, 25, 50, and 500 μM $CaCl_2$ for reference. Final concentrations were as follows: 0.5 μM rRic-8A-491, His-NCS-1/rRic-8A-491 complex, or His-NCS-1, 1 μM $r\Delta N31G\alpha_{i1}$, and 10 μM GTPγS in a reaction volume of 500 μl. Buffers for $Ca^{2+}$ concentrations of 1 μM or less were prepared with HPLC-grade water (RPI). For each assay a minimum of six technical repeats were performed. Progress curves were fit to a single or double exponential rate model using GraphPad Prism (*GraphPad Prism, 2023*). GTP binding rates of $r\Delta N31G\alpha_{i1}$ in the presence or absence of His-NCS-1/rRic-8A-491 were represented vs $CaCl_2$ concentration. Data were fit to a one-site-total binding velocity model, $v=v_o + v_{max} [Ca^{2+}]/(K_a +[Ca^{2+}])$ using GraphPad Prism (*GraphPad Prism, 2023*) to estimate an apparent $Ca^{2+}$ activation constant ($K_a$; mean ± SEM).

## Acknowledgements

MJS-B would like to thank ALBA (XALOC beamline) and ESRF synchrotrons for access and support of the staff, the mass spectrometry service from Institute 'Blas Cabrera' and Prof. Armando Albert and

Prof. Alberto Ferrús for critical revision of the manuscript. The proteomic analysis was performed in the proteomics facility of 'Centro Nacional de Biotecnología' (CSIC). SEC-MALS experiments were performed at the Spectroscopy and NMR Unit (CNIO) with the assistance of Clara M Santiveri and Ramón Campos-Olivas. This work was funded by grants from Spanish Ministry of Science and Innovation PID2019-111737RB-I00 (to MJS-B), PID2019-106608RB-I00 and PDC2022-133775-I00 (to AM), RTI2018-099985-B-I00 (to MM) and PID2020-113359GA-I00 (to JG-N). MM was supported also by CIBER of Respiratory Diseases (CIBERES) from ISCIII. AM and JG-N were supported by 'Ramón y Cajal' contracts from the Spanish Ministry of Science and Innovation (RYC-2017-22392 and RYC2018-025731-I, respectively). SA-U was funded by a PhD fellowship of the 'Diputación General de Aragón'. SP-S was supported by a contract from 'Programa de Empleo Juvenil de la Comunidad de Madrid' PEJ-2020-AI/BMD-18666. SRS would like to acknowledge NIH grant P30GM140963 for support of the Center for Biomolecular Structure and Dynamics Integrated Structural Biology Core at the University of Montana, a Pilot Project grant to LJM, and R01GM105993 to SRS.

# Additional information

## Funding

| Funder | Grant reference number | Author |
|---|---|---|
| Spanish National Plan for Scientific and Technical Research and Innovation | PID2019-111737RB-I00 | Maria Jose Sanchez-Barrena |
| Spanish National Plan for Scientific and Technical Research and Innovation | PID2019-106608RB-I00 | Alicia Mansilla |
| Spanish National Plan for Scientific and Technical Research and Innovation | PDC2022-133775-I00 | Alicia Mansilla |
| Spanish National Plan for Scientific and Technical Research and Innovation | RTI2018-099985-B-I00 | Margarita Menendez |
| Spanish National Plan for Scientific and Technical Research and Innovation | PID2020-113359GA-I00 | Javier García-Nafría |
| Spanish National Plan for Scientific and Technical Research and Innovation | RYC-2017-22392 | Alicia Mansilla |
| Spanish National Plan for Scientific and Technical Research and Innovation | RYC2018-025731-I | Javier García-Nafría |
| National Institutes of Health | P30GM140963 | Stephen Sprang |
| National Institutes of Health | R01GM105993 | Stephen Sprang |

The funders had no role in study design, data collection and interpretation, or the decision to submit the work for publication.

## Author contributions

Daniel Muñoz-Reyes, Investigation, Writing – review and editing, D M-R performed protein purification, protein complex assemblies, biochemical and biophysical experiments, protein crystallization, X-ray diffraction experiments and structure resolution; Levi J McClelland, Funding acquisition, Investigation, Writing – review and editing, L.J.M. purified the proteins needed for functional assays and performed the experiments. Also designed and analyzed functional assays; Sandra Arroyo-Urea, Investigation, Writing – review and editing; Sonia Sánchez-Yepes, Investigation, S S-Y performed co-IPs; Juan Sabín, Formal analysis, J S analyzed and interpreted ITC data; Sara Pérez-Suárez, Investigation, S

P-S performed cloning, protein expression and purification; Margarita Menendez, Conceptualization, Formal analysis, Funding acquisition, Investigation, Writing – review and editing, M M performed ITC experiments, analyzed and interpreted ITC data. M M performed the joined analysis of biophysical and functional assays; Alicia Mansilla, Formal analysis, Funding acquisition, Investigation, Writing – review and editing, A M designed the co-IPs; Javier García-Nafría, Formal analysis, Funding acquisition, Investigation, Writing – review and editing; Stephen Sprang, Formal analysis, Funding acquisition, Investigation, Writing – review and editing, S R S designed and analyzed functional assays; Maria Jose Sanchez-Barrena, Conceptualization, Formal analysis, Supervision, Funding acquisition, Investigation, Writing – original draft, Writing – review and editing, M J S -B contributed with the conception, design of the study, data analysis and interpretation. Performed the joined analysis of biophysical, functional assays and structural data, M J S-B wrote the manuscript with contributions from all authors

**Author ORCIDs**
Maria Jose Sanchez-Barrena ⬨ https://orcid.org/0000-0002-5986-1804

**Decision letter and Author response**
Decision letter https://doi.org/10.7554/eLife.86151.sa1
Author response https://doi.org/10.7554/eLife.86151.sa2

## Additional files

### Supplementary files
• Transparent reporting form

### Data availability
The atomic coordinates and structure factors have been deposited in the Protein Data Bank (https://www.pdb.org/) with codes: Structure 1 (8ALH), Structure 2 (8AHY), Structure 3 (8ALM). All data generated or analysed during this study are included in the manuscript and supporting files. Source data files have been provided for Figures 2–5.

The following datasets were generated:

| Author(s) | Year | Dataset title | Dataset URL | Database and Identifier |
|---|---|---|---|---|
| Munoz-Reyes D, Sanchez-Barrena MJ | 2023 | X-ray structure of human NCS-1 bound to Ric-8A | https://www.rcsb.org/structure/8ALH | RCSB Protein Data Bank, 8ALH |
| Munoz-Reyes D, Sanchez-Barrena MJ | 2023 | X-ray structure of human NCS-1 bound to Ric-8A | https://www.rcsb.org/structure/8AHY | RCSB Protein Data Bank, 8AHY |
| Munoz-Reyes D, Sanchez-Barrena MJ | 2023 | X-ray structure of human NCS-1 bound to Ric-8A | https://www.rcsb.org/structure/8ALM | RCSB Protein Data Bank, 8ALM |

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

# Appendix 1

**Appendix 1—key resources table**

| Reagent type (species) or resource | Designation | Source or reference | Identifiers | Additional information |
|---|---|---|---|---|
| Strain, strain background (*Escherichia coli*) | BL21* | Invitrogen | Cat # C601003 | |
| Strain, strain background (*E. coli*) | BL21(DE3) pLysS | Novagen | Cat # 70236 | |
| Strain, strain background (*E. coli*) | Rosetta2 pLysS | Novagen | Cat # 71403 | |
| Strain, strain background (*E. coli*) | WK6 cells | *McClelland et al., 2020* | Cat # 47078 | |
| Strain, strain background (*E. coli*) | BL21 CodonPlus (DE3) RIPL | *McClelland et al., 2020* | Cat # 230280 | |
| Cell line (human) | HEK293 | ATCC | Cat # CRL-1573 | |
| Antibody | Mouse monoclonal anti-V5 | Invitrogen | Cat # R960-25 | 1:5000 |
| Antibody | Rabbit polyclonal anti-NCS-1 | Cell Signaling | Cat # 8237S | 1:2000 in WB 1:500 in IP |
| Antibody | Anti-mouse antibody TrueBlot secondary | Rockland | Cat # 18-8817-30 | 1:5000 |
| Recombinant DNA reagent | Human NCS-1 (full-length) in pETDuet vector | *Canal-Martín et al., 2019* | N/A | |
| Recombinant DNA reagent | Human NCS-1ΔH10 in pETDuet vector | This work | N/A | Stop codon after residue P177 for NCS-1ΔH10 construct |
| Recombinant DNA reagent | Human His(6)-NCS-1 in pET28a+vector | This work | N/A | His-tagged NCS-1 version in pET28a+ vector |
| Recombinant DNA reagent | Human His(6)-NCS-1 (full-length) in pETDuet vector | This work | N/A | His-tagged NCS-1 version in pETDuet vector |
| Recombinant DNA reagent | Rat His(6)-Ric-8A(1-452) in pET28a vector | *Thomas et al., 2011* | N/A | |
| Recombinant DNA reagent | Rat His(6)-Ric-8A(1-423) in pET28a vector | This work | N/A | Ric-8A(1-423) truncated version of Rat His(6)-Ric-8A(1-452) in pET28a vector |
| Recombinant DNA reagent | Rat His(6)-Ric-8A(1-432) in pET28a vector | This work | N/A | Ric-8A(1-432) truncated version of Rat His(6)-Ric-8A(1-452) in pET28a vector |
| Recombinant DNA reagent | Rat His(6)-Ric-8A(1-491) in pET28a vector | *Thomas et al., 2011* | N/A | |
| Recombinant DNA reagent | Rat GST-ΔN31Gα in a pDest15 vector | *McClelland et al., 2020* | N/A | |
| Recombinant DNA reagent | Human Ric-8A deletion construct ending at G424 (hRic-8A-424) in nV5-pCDNA3.1 plasmid | This work | N/A | hRic-8A-G424 construct version of human Ric-8A in nV5-pCDNA3.1 plasmid |
| Recombinant DNA reagent | Human Ric-8A deletion construct ending at G433 (hRic-8A-433) in nV5-pCDNA3.1 plasmid | This work | N/A | hRic-8A-G433 construct version of human Ric-8A in nV5-pCDNA3.1 plasmid |
| Recombinant DNA reagent | Human Ric-8A full-length mutant (S436A, T441A) in nV5-pCDNA3.1 plasmid | This work | N/A | Phosphorylation mutant version |
| Recombinant DNA reagent | Human Ric-8A in nV5-pCDNA3.1 plasmid | *Mansilla et al., 2017* | N/A | |

*Appendix 1 Continued on next page*

*Appendix 1 Continued*

| Reagent type (species) or resource | Designation | Source or reference | Identifiers | Additional information |
|---|---|---|---|---|
| Recombinant DNA reagent | Human Ric-8A mutant (T411A, Y413A, N415A) in nV5-pCDNA3.1 plasmid | This work | N/A | T411A, Y413A, N415A mutant version of human Ric-8A in nV5-pCDNA3.1 plasmid |
| Recombinant DNA reagent | Human Ric-8A mutant (L425A, M426A) in nV5-pCDNA3.1 plasmid | This work | N/A | L425A, M426A, mutant version of human Ric-8A in nV5-pCDNA3.1 plasmid |
| Recombinant DNA reagent | Human NCS-1 in pCDNA3.1 plasmid in pCDNA3.1 plasmid | *Mansilla et al., 2017* | N/A | |
| Recombinant DNA reagent | Human NCS-1 mutant (D37A, Y52A) in pCDNA3.1 plasmid | This work | N/A | D37A, Y52A mutant version of human NCS-1 in pCDNA3.1 plasmid in pCDNA3.1 plasmid |
| Recombinant DNA reagent | Human NCS-1 mutant (R148A, R151A) in pCDNA3.1 plasmid | This work | N/A | R148A, R151A mutant version of human NCS-1 in pCDNA3.1 plasmid in pCDNA3.1 plasmid |
| Recombinant DNA reagent | Human NCS-1 mutant (D37A, R148A, R151A) in pCDNA3.1 plasmid | This work | N/A | D37A, R148A mutant version of human NCS-1 in pCDNA3.1 plasmid in pCDNA3.1 plasmid |
| Recombinant DNA reagent | Human NCS-1 mutant (D37A, Y52A, R148A, R151A) in pCDNA3.1 plasmid | This work | N/A | D37A, Y52A, R148A, R151A mutant version of human NCS-1 in pCDNA3.1 plasmid in pCDNA3.1 plasmid |
| Recombinant DNA reagent | Human NCS-1 mutant (W30A) in pCDNA3.1 plasmid | This work | N/A | W30 mutant version of human NCS-1 in pCDNA3.1 plasmid in pCDNA3.1 plasmid |
| Peptide, recombinant protein | Casein kinase II | New England Biolabs | Cat # P6010L | |
| Peptide, recombinant protein | Ric-8A P1 peptide (400-423) | GenicBio | N/A | |
| Peptide, recombinant protein | Ric-8A P2 peptide (400-429) | GenicBio | N/A | |
| Peptide, recombinant protein | Ric-8A P3 peptide (400-432) | GenicBio | N/A | |
| Chemical compound, drug | ATP | New England Biolabs | Cat # P0756S | |
| Chemical compound, drug | Water for UHPLC-MS LiChrosolv | Merck | Cat # 1037282002 | |
| Chemical compound, drug | Guanosine 5'-[g-thio]triphosphate | Sigma | Cat # G8634-10MG | |
| Software, algorithm | ImageJ | *Schneider et al., 2012* | https://imagej.net/software/imagej/ | |
| Software, algorithm | GraphPad Prism | GraphPad Software, Inc, USA; *GraphPad Prism, 2023* | https://www.graphpad.com/features | |
| Software, algorithm | AutoPROC | *Vonrhein et al., 2011* | https://www.globalphasing.com/autoproc/manual/autoPROC1.html | v1.1.7 |
| Software, algorithm | Phaser | *McCoy et al., 2007* | https://www.ccp4.ac.uk/html/phaser.html | v2.7.0 |
| Software, algorithm | Phenix | *Adams et al., 2010* | https://www.phenix-online.org/ | v1.19.2_4158 |

*Appendix 1 Continued on next page*

*Appendix 1 Continued*

| Reagent type (species) or resource | Designation | Source or reference | Identifiers | Additional information |
|---|---|---|---|---|
| Software, algorithm | COOT | *Emsley and Cowtan, 2004* | https://www2.mrc-lmb.cam.ac.uk/personal/pemsley/coot/ | 0.9.8 |
| Software, algorithm | Molprobity | *Williams et al., 2018* | http://molprobity.biochem.duke.edu/ | v4.5.2 |
| Software, algorithm | CCP4 | *Winn et al., 2011* | https://www.ccp4.ac.uk/ | v8.0 |
| Software, algorithm | PISA Server | *Krissinel and Henrick, 2007* | https://www.ebi.ac.uk/pdbe/pisa/ | 1.48 |
| Software, algorithm | PyMol | *Schrödinger, 2015* | https://pymol.org/2/ | v1.8.6.0 |
| Software, algorithm | Tycho NT.6 software | NanoTemper | https://nanotempertech.com | Tycho |
| Software, algorithm | Mascot Server | Matrix Science | https://www.matrixscience.com/ | |
| Software, algorithm | KaleidaGraph Data Analysis Program | Synergy Software | https://www.synergy.com/ | |
| Software, algorithm | AFFINImeter | AFFINImeter | https://www.affinimeter.com | |
| Other | Atomic Coordinates and Structure Factors NCS-1/Ric-8A-P2 complex | PDB | Structure 1 | 8ALH |
| Other | Atomic Coordinates and Structure Factors NCS-1/Ric-8A-P3 complex | PDB | Structure 2 | 8AHY |
| Other | Atomic Coordinates and Structure Factors NCS-1/Ric-8A-P3 complex | PDB | Structure 3 | 8ALM |
| Other | Hi Trap Phenyl FF hydrophobic column | Cytiva | 17519301 | Purification column |
| Other | Anion exchange HP Q column | Cytiva | 17115301 | Purification column |
| Other | Nickel-affinity column, HisTrap FF | Cytiva | 17525501 | Purification column |
| Other | Ni$^{2+}$-chelated Sepharose HP beads | Cytiva | 17526801 | Purification column |
| Other | HiLoad 16/600 Superdex 200 pg | Cytiva | 28989335 | Purification column |
| Other | Source 15Q column | Cytiva | 17094701 | Purification column |
| Other | Superdex 200 HR 10/300 column | Cytiva | GE17-5175-01 | Purification column |
| Other | Tycho NT.6 instrument | NanoTemper | https://nanotempertech.com/tycho/ | Nano-DSF equipment |
| Other | Tycho capillaries | NanoTemper | Cat # TY-C001 | Nano-DSF capillaries |
| Other | VP-ITC microcalorimeter | GE Healthcare | https://www.malvernpanalytical.com/en/products/product-range/microcal-range | ITC equipment |
| Other | BLItz system | ForteBio | BLItz from ForteBio | BLI equipment |
| Other | Ni-NTA biosensors | Sartorius | CA89413-836 | BLI biosensors |
| Other | Protein-G-Sepharose | Sigma-Aldrich | CAT # P3296-1ML | Antibodies purification |

