## [Editor Report]

This work provides a comprehensive set of convincing biochemical and structural experiments to determine the molecular basis of calcium-sensitive regulation of the guanine exchange factor Ric8A by the neuronal calcium sensor 1 (NCS-1). The Ric-8A/NCS-1 interface is a promising target for modulation of synaptic activity under pathological conditions, and this work will have important implications for scientists interested in G-protein signaling and molecular interactions that contribute to synapse function.

---

## [Decision Letter]

**Decision letter after peer review:**

Thank you for submitting your article "The neuronal calcium sensor NCS^-1^ regulates the phosphorylation state and activity of the Gα chaperone and GEF Ric-8A" for consideration by *eLife*. Your article has been reviewed by 3 peer reviewers, including Randy B Stockbridge as Reviewing Editor and Reviewer #3, and the evaluation has been overseen by Richard Aldrich as the Senior Editor. The following individual involved in the review of your submission has agreed to reveal their identity: Jennifer Cash (Reviewer #2);

Essential revisions:

1. The reviewers do not think that the cryo-EM maps, at the current resolution, support the conclusions drawn from this data including the model for the full assembly. The reviewers encourage the authors to continue to improve the protein prep and grid preparation to achieve higher resolution. However, the reviewers also acknowledge that this may be difficult. If the authors are unable to improve the maps, the conclusions and models based on cryo-EM data should be removed from the manuscript.

The reviewers were also concerned that the nanobodies may interfere with the native binding interface. If the cryo-EM maps are improved and included, the authors should provide evidence that the nanobodies do not exert such an influence.

2. Since the structures of the Ric-8A/NCS-1 interface are inferred from Ric-8A peptides, it is critical to validate this interface with mutational analysis of the full-length or truncated Ric-8A.

3. Additional control or replicate experiments should be reported, including:

– Measurements of the Gα GTPγS binding rates in the presence of NCS-1 alone, as well as the data points for 1 and 50 μm Ca^2+^ across all conditions.

– The data for NSC-1/Ric-8A at all Ca^2+^ concentrations are compared to the 0 Ca^2+^ condition for Gα (Figure 8A). The comparison should be between the data points with the same Ca^2+^ concentration.

– Figure 2B and 2D: Please provide replicate measurements and errors to assess reproducibility, including replicates from at least independent protein preps. The authors should provide an explanation of what the points on these plots indicate, and how the inflection values were determined.

– The section on Ric-8A phosphorylation should include additional positive controls to demonstrate that the casein kinase treatment successfully phosphorylates Ric-8A, such as the LC-MS/MS experiment.

4. The authors should clarify the conclusions drawn from the calcium dependence of GEF activity, supplementing with additional experiments if necessary. In Figure 8B, the effect of ca^2+^ on the GTPγS binding by Gα alone should be considered, since this would correspondingly diminish the rescue effect of Ca^2+^ on the GEF activity of Ric-8A in the presence of NCS-1. The authors should also address why the rescue occurs at high concentrations of Ca^2+^. The authors hypothesize that the rescue occurs because NSC-1 no longer effectively binds to Ric-8A at high Ca^2+^. However, the authors should comment on why the fast component of the Ric-8A-stimulated GTPgS binding is not restored at high Ca^2+^ (Figure 8-suppl Figure 1). Finally, the authors should comment on why Ca^2+^ binding affinities for the three NCS-1 binding sites (Kd values < 1 uM, Figure 4D, Table 2) do not match the effects of Ca^2+^ in Figure 8, which require up to 500 μm Ca^2+^.

5. The large increase in the thermostability of Ric-8A in the complex with NCS-1 (Figure 2) seems to disagree with the proposed detachment of the Ric-8A arm/heat repeats 8 and 9 from the core of Ric-8A. This should be considered and discussed.

6. Please provide additional discussion and rationale for the following experiments:

– Nano-DSF experiments (starting line 180): Here (and elsewhere in the manuscript) the reader would benefit from a short description of the assay and analysis.

– Tryptophan emission fluorescence experiments: Please describe this assay and why it can be utilized for these proteins. Where are the tryptophans located, and what does the data tell us?

– Clarification is needed on ITC data interpretations, including how the authors determine the sequence of calcium binding to each site.

*Reviewer #1 (Recommendations for the authors):*

1. It is odd that the intensity of the NCS-1 band in the SEC-purified Ric-8A/NCS-1 complex is roughly double that for Ric-8A when the opposite would be expected based on the MW of the proteins (Figure 2A).

2. The large increase in the thermostability of Ric-8A in the complex with NCS-1 (Figure 2) is in stark contrast with the proposed detachment of the Ric-8A arm/heat repeats 8 and 9 from the core of Ric-8A.

3. Many of the NCS-1 contact residues of Ric-8A from the structures with peptides are involved in the intramolecular interactions and are not available for binding NCS-1 (Figure 3). What residues of Ric-8A are involved in the initial recognition of NCS-1? The interface from the "peptide" structures needs to be validated with mutational analysis of Ric-8A, which may identify residues involved in the initial recognition.

4. The Ca^2+^ binding affinities for the three NCS-1 binding sites (Kd values < 1 uM, Figure 4D, Table 2) do not match the effects of Ca^2+^ in Figure 8 requiring up to 500 μm Ca^2+^.

5. The resolution of the cryo-EM reconstruction is low, and the maps are visually not improving through multiple rounds of 3D classification (Figure 4S). With significant portions falling out of the density, the Ric-8A nanobodies do not anchor Ric-8A into the map well. More importantly, the placement of NCS-1 into the map does not appear to have a structural rationale. What force pulls NCS-1 with Ric-8A repeats 8 and 9 away from the arm/heat core domain and towards the N-terminal part of Ric-8A (and keeps it stable in that position)? From the model, this could only be a nonspecific interaction of NCS-1 with the Nb8119 nanobody. Optimization of the sample/grid preparation is required for unequivocal cryo-EM reconstruction of the complex structure.

6. The analysis of the effect of NCS-1 and Ca^2+^ on the GEF activity of Ric-8A lacks important controls and comparisons. Missing are the measurements of the Gα GTPγS binding rates in the presence of NCS-1 alone, as well as the data points for 1 and 50 μm Ca^2+^ across all conditions. Also, the data for NCS-1/Ric-8A at all Ca^2+^ concentrations are compared to the 0 Ca^2+^ condition for Gα (Figure 8A). The comparison should be between the data points with the same Ca^2+^ concentration. Similarly, the meaning of the plot in Figure 8B is unclear because the effect of Ca^2+^ on the GTPγS binding by Ga alone is not taken into account. If one takes into account the activating effect of Ca^2+^ on Gα, the rescue effect of Ca^2+^ on the GEF activity of Ric-8A in the presence of NCS-1 is not as apparent. Why does the rescue occur at high concentrations of Ca^2+^ (see comment 4)? It is hypothesized that the rescue occurs because at high Ca^2+^ NSC-1 no longer effectively binds to Ric-8A. Why then is the fast component of the Ric-8A-stimulated GTPγS binding not restored at high Ca^2+^ (Figure 8-suppl Figure 1)?

*Reviewer #2 (Recommendations for the authors):*

– Nano-DSF experiments (starting line 180): Here (and elsewhere in the manuscript) the reader would benefit from a short description of the assay being used before describing the results. What is a "Ti" value? In Figure 2B, why are there two points on this curve, and how were they determined? The inflection points are not convincing, and there are no statistics presented to allow the reader to assess the reproducibility and significance of the data. Data shown in Figure 2D is equally unconvincing.

– Tryptophan emission fluorescence experiments: Please describe this assay and why it can be utilized for these proteins. Where are the tryptophans located, and what does the data tell us?

– Results in the paragraph starting line 341: How are the authors determining the sequence of calcium binding to each site? How are the data correlated to each binding site? Clarification is needed on ITC data interpretations here.

– The major concern of this reviewer is in the cryo-EM data presented. A very low-pass filtered map is presented (17 Å), and it is stated that low resolution is due to high levels of dissociation and conformational heterogeneity. How was this determined? From the micrograph, it looks like the particles are buried in thick ice, which would limit the resolution. From the 2D class averages, the complete particles look mostly intact. Resolution is likely also limited by a low number of particles representing the final map. The authors state that the model could be "unambiguously fit using the Nbs as fiducial markers" (line 370). From the maps presented, I do not see this, and I would argue that nothing can be unambiguously fit into a cryo-EM map at 17 Å. All three nanobodies are protruding from the map as well as bits of Ric-8A. Additionally, there are unaccounted-for parts of the map that are left unmodeled. The cryo-EM data are of poor quality, and the modeling is not convincing. Furthermore, the conclusions drawn from this data add little to the story being presented in the manuscript. I recommend removing this data from the manuscript, or, at the very least, moving it entirely to the supplemental data section. This would not diminish the impact of the manuscript.

– Abstract: Briefly define what Neuronal Calcium Sensor 1 is – what type of protein is this?

– Figure 7: This is out of order with the flow of the presented figures. Please fix.

*Reviewer #3 (Recommendations for the authors):*

1. The section on Ric-8A phosphorylation should include additional positive controls to demonstrate that the casein kinase treatment successfully phosphorylates Ric-8A, such as the LC-MS/MS experiment. This would both bolster the conclusion from LC-MS/MS that the complexed Ric8A resists phosphorylation, and provide support for the statement that uncomplexed Ric8A is fully phosphorylated (line 409), which is difficult to conclude from ion exchange chromatography alone since this is a low-resolution method (although it is clear that the protein is at least partially phosphorylated).

2. What evidence is there that the nanobodies don't interfere with the native binding interaction between Ric8A and NCS-1? Based on the figures, 8109 looks particularly problematic in terms of being close to the proposed interface, as if it might cause the proposed hinge to extend in a non-native way. It would strengthen this section to perform a binding experiment in the presence of the nanobody.

3. Can the authors quantify their fits to the density envelope in cryo-EM? (perhaps by determining RSCC values?) The authors state that they identified the proper docking configuration because the nanobodies fit the density better, but even in the favored interpretation, nanobody 8109 and 8117 seem to be sticking out of the density envelope. It's hard to tell by eye that one version in figure S5 is clearly better than the others.

[Editors’ note: further revisions were suggested prior to acceptance, as described below.]

Thank you for resubmitting your work entitled "The neuronal calcium sensor NCS^-1^ regulates the phosphorylation state and activity of the Gα chaperone and GEF Ric-8A" for further consideration by *eLife*. Your revised article has been evaluated by Kenton Swartz (Senior Editor) and a Reviewing Editor.

The manuscript has been improved, and the reviewers are satisfied with most of the changes, but there is one remaining issue that needs to be addressed, as outlined below:

Two reviewers with expertise in cryo-EM both have reservations about the updated cryo-EM data processing and remain concerned about the model fitting to the map. However, the reviewers think that the manuscript still provides new, valuable insight even without the cryo-EM data and that this data is not essential to support the main conclusions. Thus, the cryo-EM data should be removed from the manuscript, and the conclusions and models should be rewritten accordingly. The full reviews are below.

*Reviewer #1 (Recommendations for the authors):*

I would still recommend that the authors remove the cryo-EM analysis from the manuscript. The new analysis of the old data does not seem to significantly improve the reconstruction. The use of the previous cryo-EM map to search for particles may have introduced model bias, and it may have led to overfitting the noise. This may explain why the new 2D classes are streaky. In my view, the model fitting into the map remains ambiguous.

*Reviewer #2 (Recommendations for the authors):*

Overall, the authors have assembled a thorough and satisfactory response to most of the requests of the reviewers.

However, after reviewing the revised data, my concerns on the presented cryo-EM data still stand. In fact, I am now more confused about the way the model of the complex has been assembled into the map. It appears that NCS-1 is no longer making contact with Ric-8A. Although the 2D classification images show more detail, they also show spikey features, likely due to the alignment of noise. I disagree that the new map is more informative. The presented micrograph is also difficult to interpret, and I'm not sure what is meant to be seen here. It seems that the biochemistry of the sample needs to be revisited and not further data collection and processing of the current sample. I don't support publishing this cryo-EM data. Again, I don't think these data are needed to support the main conclusions of the paper, and the paper would not carry any less weight without them.

---

## [Author Response]

Essential revisions:1. The reviewers do not think that the cryo-EM maps, at the current resolution, support the conclusions drawn from this data including the model for the full assembly. The reviewers encourage the authors to continue to improve the protein prep and grid preparation to achieve higher resolution. However, the reviewers also acknowledge that this may be difficult. If the authors are unable to improve the maps, the conclusions and models based on cryo-EM data should be removed from the manuscript.The reviewers were also concerned that the nanobodies may interfere with the native binding interface. If the cryo-EM maps are improved and included, the authors should provide evidence that the nanobodies do not exert such an influence.

We acknowledge that the heterogeneity of the sample limits the conclusions that can be extracted from the cryo-EM reconstruction. For this reason, we have made significant additional efforts to improve the current cryo-EM map while also tuning down the conclusions extracted from the model:

a) We have made new NCS-1ΔH10/Ric-8A-452 complex sample (with two Nbs Nb8117 and Nb8119) and vitrified the complex in cryo-grids with carbon support, attempting to remove potential effects of the air-water interface that could trigger the disassembly of the complex. We then collected >30,000 micrographs however, without success. We saw an even higher disassembly, likely due to the lower concentration used for carbon support grid (3-fold lower than what we use for freestanding ice). Hence, this approach failed to improve the sample quality. Although we did not manage to obtain 3D maps, 2D averages were similar to those previously achieved and those in the improved dataset, albeit with worse quality.

b) We have also used both Relion and Cryosparc to reprocess all four data sets that had been collected on the NCS-1ΔH10/Ric-8A-452 complex with three nanobodies. We had success by using the final cryo-EM map presented in the previous manuscript to search for particles in previous datasets. Using this strategy, we managed to achieve improved data in one of the earlier datasets, as seen by the 2D averages and cryo-EM map (Author response image 1 the comparison in Figure 1). Although still at low resolution (10 Å), it is more homogenous and density can be assigned with no ambiguity (i.e. nanobodies fit clearly into their own density when fitting the Ric-8A and 3Nbs *en bloc*). We show in Author response image 1 a comparison of the different views of the old and new map with the fit. A quantification of the RSCC of the old and new maps shows a significant improvement (from a CC of 0.64 to 0.75).

**Author response image 1. sa2fig1:** Comparison of 2D averages and cryo-EM map of the previously presented model and the currently improved one. (Top left) 2D averages from the previous and current models, generated with the final particles after 3D classification. The old and new cryo-EM maps with docked PDBs from a front (top right), side (bottom left) and top (bottom right) views. The cryo-EM map is displayed as a grey transparent surface with Ric-8A (red cartoon), NCS^-1^ (purple cartoon) and the three nanobodies (yellow, cyan and green cartoons) docked into it.

Additionally, the reviewers were concerned that the nanobodies may interfere with the native binding interface*,* specifically with Nb8109, which is close to the NCS-1/Ric-8A interface*.* We would like to note that we assemble the NCS-1/Ric-8A first and then add the three nanobodies, showing that the binding of NCS1 to the complex does not preclude Nb binding (SEC-MALS results estimate the molecular mass of a homogeneous heteropentameric complex, therefore, the Ric-8A regions devoted to Nb recognition are available in the NCS-1/Ric-8A complex). Our assembly procedure is the same as that used in the Ric8A/Gα/3Nb complex (McClelland et al., 2020). The structure of Ric-8A/Gα has been solved also without nanobodies (Seven et al. 2020) and the structural comparison of both structures shows that they are virtually the same, with local changes in large loops that are implicated in Nb recognition. It has been described that the ARM-HEAT repeat domain is arranged in three subdomains constituted by repeats 1-4, 5-6 and 7-9 (Zeng et al., 2019), and that these subdomains can suffer quasi-rigid body angular displacements, contributing to the generation of different curvatures of the ARM-HEAT repeat domain. Since Nb8119 binds to repeats 1-2, Nb8117 to repeat 5-6 and Nb8109 to repeat 7, all at the middle of the subdomain and not to the junctions, we don’t feel that they interfere with mobility. Additionally, Nbs do not bind to the repeat 9, which physically contacts NCS-1.

Overall, we feel that the cryo-EM map, although at low resolution, makes some relevant contributions: (a) it supports the detachment of Ric-8A repeat 9 from the ARM-HEAT repeat domain, as suggested by the crystal structure of NCS-1/Ric-8A peptide complexes; (b) it highlights the different nature of the Ric-8A/NCS-1 complex when compared to Ric-8A/Gα (while the latter has three strong interfaces, NCS-1 interacts with the C-terminus of Ric-8A ARM-HEAT repeat domain, through a flexible region and makes no interactions with the core of the domain); and (c) it highlights that therapeutic modulation of Ric-8A/NCS-1 interaction would exclusively require regulation of the interaction observed in the crystal structure. Therefore, we believe that keeping these data in the manuscript is positive.

We have introduced the corresponding changes in the Results (starting at line 423), Discussion (line 565) and Materials and methods (line 1046) sections, as well as in Figure 5, Figure 5-supplement 1 and 2, and Figure 8C and 8D (previously Figure 7C and 7D). In this new version of the manuscript, and thanks to one of the reviewers´ comments, we have properly named Figure 7. Taking the flow of figures along the paper, Figure 8 should be named Figure 7, and the other way around, Figure 7 should be named Figure 8.

2. Since the structures of the Ric-8A/NCS-1 interface are inferred from Ric-8A peptides, it is critical to validate this interface with mutational analysis of the full-length or truncated Ric-8A.

We have designed several human hRic-8A and hNCS-1 full-length mutants and conducted co-immunoprecipitations (experiments in triplicate) to validate the protein-protein interface observed in the crystal structures of NCS-1 bound to Ric-8A peptides (see new Figure 3D and Table 3). Concretely, we have tested with a total of 4 NCS-1 mutants the relevance of disrupting H-bonds between NCS-1 and Ric8A. Also, we have tested the relevance of interactions that involve the Ric-8A R1-R2 loop and the C-terminal region of helix R2, which acquires an extended coil structure when bound to Ga and is helical in the presence of NCS-1. The analysis of the results validates the crystal structure presented and also gives insights on how the initial recognition of Ric-8A may take place. Please, see the new data in the following sections: Results (starting at line 295), Discussion (line 539 and new paragraph starting at line 587) and Materials and methods (starting at line 1019 and end of paragraph). Also, the new Figure 3D, Table 3 (line 1144) and changes in Figure 3-Supplement 2, where mutated residues are indicated.

3. Additional control or replicate experiments should be reported, including:– Measurements of the Gα GTPγS binding rates in the presence of NCS-1 alone, as well as the data points for 1 and 50 μm Ca^2+^ across all conditions

We have performed the suggested experiments and tested also lower Ca^2+^ concentrations, which are physiologically more relevant. Please, see the new data in the Results and Methods sections related to nucleotide exchange experiments (starting at line 494 and line 1079, respectively) and new Figure 7 (previously Figure 8) and Figure 7-supplementary figure 2.

– The data for NSC-1/Ric-8A at all Ca^2+^ concentrations are compared to the 0 Ca^2+^ condition for Gα (Figure 8A). The comparison should be between the data points with the same Ca^2+^ concentration.

We have performed the suggested experiments and compared data points with the same Ca^2+^ concentration. Please, see the new data in the Results and Methods sections related to nucleotide exchange experiments (starting at line 494 and line 1079, respectively) and the new figures: Figure 7B and Figure 7-supplementary figure 2.

– Figure 2B and 2D: Please provide replicate measurements and errors to assess reproducibility, including replicates from at least independent protein preps. The authors should provide an explanation of what the points on these plots indicate, and how the inflection values were determined.

We have performed three independent replicates for each curve, providing a mean and SEM value of each inflection temperature, Ti (see new Figure 2B and 2D and corresponding figure legends). We have added details on the technique and how the inflection values (Ti, dots on plots) are determined in the Materials and methods section (starting at line 863).

– The section on Ric-8A phosphorylation should include additional positive controls to demonstrate that the casein kinase treatment successfully phosphorylates Ric-8A, such as the LC-MS/MS experiment.

We now supply an additional positive control that demonstrates that under the experimental conditions, CK2 phosphorylates Ric-8A. See please the new Figure 6-supplement figure 1, Results section (see line 469) and finally Methods section (line 891).

We have evaluated the phosphorylation state of rRic-8A-452 (control experiment) by MS/MS fragmentation (see new Figure 6-supplement figure 1). We have detected two phosphopeptides with high scores in which S435 is phosphorylated and M425 is oxidized, GLoxMAGGRPEGQYpSE and GLoxMAGGRPEGQYpSEDEDTDTEEYR. The identification of two different peptides with the same phosphorylation site provides confidence in the result.

Unfortunately, we were not able to detect phosphorylation of T440. It would be due to two facts:

1. CK2 mostly phosphorylates serine and rarely threonine see CK2 vendor information (https://international.neb.com/products/p6010-casein-kinase-ii-ck2#Product%20Information_Properties%20&%20Usage) and an excess of enzyme is needed to achieve phosphorylation of T440 PapasergiScott et al. (2018), McClelland et al. (2020). Reaction temperatures were set to 37ºC in Papasergi-Scott et al. (2018) and 25ºC in McClelland et al. (2020), while in our experiment it was 18ºC. The lower temperature used may have been detrimental to T440 phosphorylation.

2. The mass/charge (M/z) value of the S435,T440 di-phosphorylated peptide (M/z=966) is above the good detection range (650-800 M/z) of the equipment we used, which may add difficulties in the detection.

Nevertheless, both Ric-8A-452 and the NCS^-1^/Ric-8A were subjected to phosphorylation under the same experimental conditions and no phosphorylated peptide was detected in Ric-8A in the context of the NCS1/Ric-8A complex.

4. The authors should clarify the conclusions drawn from the calcium dependence of GEF activity, supplementing with additional experiments if necessary. In Figure 8B, the effect of Ca^2+^ on the GTPγS binding by Gα alone should be considered, since this would correspondingly diminish the rescue effect of Ca^2+^ on the GEF activity of Ric-8A in the presence of NCS-1.

We have followed the reviewer´s recommendation and clarified the conclusions. See, please, Results section related to nucleotide exchange experiments (starting at line 481). After subtracting the effect of Ca^2+^ on Gα_i1_ alone, ~35 % of the nucleotide exchange rate (instead of 85%) is restored at 25 µM CaCl_2_, compared to levels observed in the presence of Ric-8A-491(new Figure 7). The recovery increased at higher cation concentrations, and an apparent constant K_a-NCS-1_ = 61 + 35 µM was estimated for the Ca^2+^ induced effect on the NCS-1/Ric-8A complex, leading to Ric-8A disassembly from NCS-1 and enabling Ric8A to catalyze Gα_i1_ nucleotide exchange. We discuss that the NCS-1/Ric-8A complex must be partially kinetically trapped. The extensive conformational changes that must occur in order to disengage the a8-b9 HEAT repeat helices of Ric-8A from NCS-1 and subsequently rearrange to accommodate binding to Gα_i1_ likely imposes a substantial kinetic barrier. Also, in vitro and in the absence of other NCS-1 targets (this is not the case in vivo, as we point out in Figure 9 and the corresponding Discussion section) the NCS-1/Ric8A complex is thermodynamically favored, since the complete hydrophobic crevice of NCS-1 is buried in the complex, while in the absence of target, it would be partially exposed to the solvent.

Additional data are also provided to better understand the interplay between NCS-1, Ric-8A and Gi. See new Figure 4E, Figure 4-supplement figure 2 and Table 4. Using biolayer interferometry (BLI) (see Results section starting at line 407 and the corresponding methods starting at line 977), we have found that the affinity of full-length NCS-1 for a Ric-8A peptide is reduced with the increase of Ca^2+^ concentrations. Our data show that when the regulatory EF-4 Ca^2+^ binding site is empty, the apparent dissociation constant of the NCS-1/Ric-8A-P3 complex is in the hundreds of µM range. However, at high cellular Ca^2+^ concentrations (250nM), NCS-1 affinity for Ric-8A-P3 is reduced 1.5 times, and at 425 nM Ca^2+^, the interaction is greatly reduced. In such situation, Ric-8A would be free to interact with Gα_i_. The new data are found in: Materials and methods (new “Biolayer Interferometry” section starting at line 977), Results (new paragraph starting at line 407), Table 4 (line 1147) and the new figures Figure 4E and Figure 4-supplement 2.

The authors should also address why the rescue occurs at high concentrations of Ca^2+^. The authors hypothesize that the rescue occurs because NSC-1 no longer effectively binds to Ric-8A at high Ca^2+^. However, the authors should comment on why the fast component of the Ric-8A-stimulated GTPγS binding is not restored at high Ca^2+^ (Figure 8-suppl Figure 1).

Please, see our arguments and discussion above. Also comments in the corresponding Results section (starting at line 407). The fact that the complex gets partly kinetically trapped would not allow the detection of the binding of GTPγS to an intermediary complex of Ric-8A with nucleotide-free ΔN31Gαi1.

Finally, the authors should comment on why Ca^2+^ binding affinities for the three NCS-1 binding sites (Kd values < 1 uM, Figure 4D, Table 2) do not match the effects of Ca^2+^ in Figure 8, which require up to 500 μm Ca^2+^.

We comment on these differences in the Results section related to the nucleotide binding assays (starting at line 516). There are several reasons that may account for the high Ca^2+^ concentrations the rescue occurs compared with the Ca^2+^ affinity of the regulatory EF-4 site. First, the experimental conditions were different in each experiment: Na^+^ concentration was 50 mM and 200 mM in the ITC and nucleotide exchange assays, respectively. Second, the nucleotide exchange assay requires 2 mM MgCl2, and Mg^2+^ competes with Ca^2+^ for binding. Third, and more importantly, the apparent activation constant K_a-NCS-1_ includes at least three different events: Na^+^ dissociation from NCS-1 EF-4 (200 mM NaCl experimental conditions), dissociation of Ric-8A from the NCS-1/Ric-8A complex, and binding of Ca^2+^ to NCS-1. Na^+^ dissociation from EF-4 and the disassembly of NCS-1 from Ric-8A would negatively contribute to the overall free energy change, thereby reducing the apparent affinity of NCS-1 for Ca^2+^. Therefore, a higher concentration of Ca^2+^ is needed to saturate the EF-4 hand in nucleotide exchange functional assays compared to ITC experiments.

5. The large increase in the thermostability of Ric-8A in the complex with NCS-1 (Figure 2) seems to disagree with the proposed detachment of the Ric-8A arm/heat repeats 8 and 9 from the core of Ric-8A. This should be considered and discussed.

As it is pointed out, there is an important improvement of Ric-8A thermostability (>20ºC) upon NCS-1 binding (Figure 2B). This result was indicative of the strong interactions established in the complex between the Ric-8A-452 and NCS-1ΔH10. It also means that the free energy of binding would be high enough as to pay the energy penalty derived from the structural changes experienced by Ric-8A to give rise to the complex with NCS-1. The impact of NCS-1 binding to Ric-8A probably goes beyond the structural rearrangement that takes place at repeat 9 and its detachment from repeat 8. In this respect, it is worth noting that global changes in the Ric-8A ARM-HEAT repeat domain have been observed when comparing the phosphorylated (pRic-8A) with the unphosphorylated version (uRic-8A) of the protein. Specifically, Zeng et al. (2019) showed that phosphorylation of Ric-8A induces a global change in rRic-8A-452 structure. The combination of the crystallographic data of pRic-8A-452 with low-resolution SAXS data of the phosphorylated and unphosphorylated proteins in solution suggested that there were quasi-rigid body angular displacements of three subdomains (repeats 1-4, 5-6 and 7-9) of the ARM-HEAT repeat domain. This rearrangement is translated into different contacts between subdomains, along with a different global shape and curvature of the ARM-HEAT repeat domain (see new Figure 8D). Finally, although the current resolution of our cryo-EM data does not allow us to describe the structural reorganization, the curvature of Ric-8A in the low-resolution uRic-8A/NCS-1 complex does not match that of the low-resolution model uRic8A or those of pRic-8A or pRic-8A/Gα complex (Figure 8D), suggesting that NCS-1 induces a global rearrangement. We now comment on this issue in the Discussion section, starting at line 565.

6. Please provide additional discussion and rationale for the following experiments:– Nano-DSF experiments (starting line 180): Here (and elsewhere in the manuscript) the reader would benefit from a short description of the assay and analysis.

Following the reviewer´s recommendation, we have included a short description of the assay and the analysis of data in the Materials and methods section (line 863).

– Tryptophan emission fluorescence experiments: Please describe this assay and why it can be utilized for these proteins. Where are the tryptophans located, and what does the data tell us?

As done with nano-DSF and BLI biophysical techniques (new biophysical data), we have included a short description of the assay and why it can be used with NCS^-1^ (see Materials and methods – line 942). We have also added in the Results section (line 367) what the data tell on why Na^+^, which binds at EF-hand EF-4, can be sensed by a Trp residue located at EF-3, W103.

– Clarification is needed on ITC data interpretations, including how the authors determine the sequence of calcium binding to each site.

As suggested by the reviewers, we have clarified ITC data. Please, see Results (line 390) and Materials and methods (line 970) sections.

Reviewer #1 (Recommendations for the authors):1. It is odd that the intensity of the NCS-1 band in the SEC-purified Ric-8A/NCS-1 complex is roughly double that for Ric-8A when the opposite would be expected based on the MW of the proteins (Figure 2A).

This type of calcium sensors run anomalously on gels since they are highly thermostable due to their high affinity for calcium. Despite the samples are boiled, they maintain a certain fold because structural Ca^2+^ binding sites are not completely destroyed. The protein is not completely denatured and runs at a lower molecular weight (NCS-1ΔH10 is 20,500 Da but clearly migrates below 20,000, see Figure 2). Also, their aberrant migration is translated to bands that sometimes look wider. We would like to point out that we have run SEC-MALS with the nanobodies assembled NCS-1/Ric-8A complex and the results point to a heteropentameric NCS-1/Ric-8A/Nb8117/Nb8119/Nb8109 (Material and methods line 817).

2. The large increase in the thermostability of Ric-8A in the complex with NCS-1 (Figure 2) is in stark contrast with the proposed detachment of the Ric-8A arm/heat repeats 8 and 9 from the core of Ric-8A.

As indicated in Essential Revision 5, we now clarify and discuss on the structural implications of the observed increase in thermostability.

3. Many of the NCS-1 contact residues of Ric-8A from the structures with peptides are involved in the intramolecular interactions and are not available for binding NCS-1 (Figure 3). What residues of Ric-8A are involved in the initial recognition of NCS-1? The interface from the "peptide" structures needs to be validated with mutational analysis of Ric-8A, which may identify residues involved in the initial recognition.

Please, see Essential revision question 2. The mutations point to the relevance of the interactions between the upper part of the NCS-1 crevice and the C-terminal half of helix R2. In the absence of NCS-1, this region adopts an extended coiled structure, and in the absence of phosphorylation, it would be detached from Ric8A ARM-HEAT repeat domain, and thus, it would be available. The presence of NCS-1 would promote the formation of the helical structure and this could constitute the first steps that would trigger the structural reorganization of Ric-8A. We now discuss this in the manuscript (Discussion section starting at line 587).

4. The Ca^2+^ binding affinities for the three NCS-1 binding sites (Kd values < 1 uM, Figure 4D, Table 2) do not match the effects of Ca^2+^ in Figure 8 requiring up to 500 μm Ca^2+^.

Discussed above at Essential revision question 4.

5. The resolution of the cryo-EM reconstruction is low, and the maps are visually not improving through multiple rounds of 3D classification (Figure 4S). With significant portions falling out of the density, the Ric-8A nanobodies do not anchor Ric-8A into the map well. More importantly, the placement of NCS-1 into the map does not appear to have a structural rationale. What force pulls NCS-1 with Ric-8A repeats 8 and 9 away from the arm/heat core domain and towards the N-terminal part of Ric-8A (and keeps it stable in that position)? From the model, this could only be a nonspecific interaction of NCS-1 with the Nb8119 nanobody. Optimization of the sample/grid preparation is required for unequivocal cryo-EM reconstruction of the complex structure.

As discussed in Essential revision question 1, the improvement of cryo-EM data processing suggests that there is no contact between Nb8119 and NCS-1.

6. The analysis of the effect of NCS-1 and Ca^2+^ on the GEF activity of Ric-8A lacks important controls and comparisons. Missing are the measurements of the Gα GTPγS binding rates in the presence of NCS-1 alone, as well as the data points for 1 and 50 μm Ca^2+^ across all conditions. Also, the data for NCS-1/Ric-8A at all Ca^2+^ concentrations are compared to the 0 Ca^2+^ condition for Gα (Figure 8A). The comparison should be between the data points with the same Ca^2+^ concentration. Similarly, the meaning of the plot in Figure 8B is unclear because the effect of Ca^2+^ on the GTPγS binding by Gα alone is not taken into account. If one takes into account the activating effect of Ca^2+^ on Gα, the rescue effect of Ca^2+^ on the GEF activity of Ric-8A in the presence of NCS-1 is not as apparent. Why does the rescue occur at high concentrations of Ca^2+^ (see comment 4)? It is hypothesized that the rescue occurs because at high Ca^2+^ NCS-1 no longer effectively binds to Ric-8A. Why then is the fast component of the Ric-8A-stimulated GTPγS binding not restored at high Ca^2+^ (Figure 8-suppl Figure 1)?

Discussed above. Please, see Essential revision question 4.

Reviewer #2 (Recommendations for the authors):– Nano-DSF experiments (starting line 180): Here (and elsewhere in the manuscript) the reader would benefit from a short description of the assay being used before describing the results. What is a "Ti" value? In figure 2B, why are there two points on this curve, and how were they determined? The inflection points are not convincing, and there are no statistics presented to allow the reader to assess the reproducibility and significance of the data. Data shown in figure 2D is equally unconvincing.

Please, see Essential revision question 3, where we address the reviewer´s concerns.

– Tryptophan emission fluorescence experiments: Please describe this assay and why it can be utilized for these proteins. Where are the tryptophans located, and what does the data tell us?

Please, see Essential revision question 6, where we address the reviewer´s concerns.

– Results in the paragraph starting line 341: How are the authors determining the sequence of calcium binding to each site? How are the data correlated to each binding site? Clarification is needed on ITC data interpretations here.

Please, see Essential revision question 6, where we address the reviewer´s concerns.

– The major concern of this reviewer is in the cryo-EM data presented. A very low-pass filtered map is presented (17 Å), and it is stated that low resolution is due to high levels of dissociation and conformational heterogeneity. How was this determined? From the micrograph, it looks like the particles are buried in thick ice, which would limit the resolution. From the 2D class averages, the complete particles look mostly intact. Resolution is likely also limited by a low number of particles representing the final map. The authors state that the model could be "unambiguously fit using the Nbs as fiducial markers" (line 370). From the maps presented, I do not see this, and I would argue that nothing can be unambiguously fit into a cryo-EM map at 17 Å. All three nanobodies are protruding from the map as well as bits of Ric-8A. Additionally, there are unaccounted-for parts of the map that are left unmodeled. The cryo-EM data are of poor quality, and the modeling is not convincing. Furthermore, the conclusions drawn from this data add little to the story being presented in the manuscript. I recommend removing this data from the manuscript, or, at the very least, moving it entirely to the supplemental data section. This would not diminish the impact of the manuscript.

We agree with the reviewer that the previous model was of low quality and have made significant improvements that we believe make relevant contributions to the manuscript (see Essential revision question 1).

– Abstract: Briefly define what Neuronal Calcium Sensor 1 is – what type of protein is this?

We have now defined what type of protein is: EF-hand Ca^2+^ binding protein.

– Figure 7: This is out of order with the flow of the presented figures. Please fix.

We have corrected this. Previous Figure 8 has been named Figure 7 and conversely, previous Figure 7 is now called Figure 8.

Reviewer #3 (Recommendations for the authors):1. The section on Ric-8A phosphorylation should include additional positive controls to demonstrate that the casein kinase treatment successfully phosphorylates Ric-8A, such as the LC-MS/MS experiment. This would both bolster the conclusion from LC-MS/MS that the complexed Ric8A resists phosphorylation, and provide support for the statement that uncomplexed Ric8A is fully phosphorylated (line 409), which is difficult to conclude from ion exchange chromatography alone since this is a low-resolution method (although it is clear that the protein is at least partially phosphorylated).

Please, see Essential revision question 6, where we address the reviewer´s concerns.

2. What evidence is there that the nanobodies don't interfere with the native binding interaction between Ric8A and NCS-1? Based on the figures, 8109 looks particularly problematic in terms of being close to the proposed interface, as if it might cause the proposed hinge to extend in a non-native way. It would strengthen this section to perform a binding experiment in the presence of the nanobody.

Please, see Essential revision question 1, where we address the reviewer´s concerns.

3. Can the authors quantify their fits to the density envelope in cryo-EM? (perhaps by determining RSCC values?) The authors state that they identified the proper docking configuration because the nanobodies fit the density better, but even in the favored interpretation, nanobody 8109 and 8117 seem to be sticking out of the density envelope. It's hard to tell by eye that one version in figure S5 is clearly better than the others.

We have used RSCC to quantify the fit of the overall model to the map, yield a CC of 0.75 (added in methods now). This is a significant improvement in CC from the previous model (0.64).

[Editors’ note: further revisions were suggested prior to acceptance, as described below.]

The manuscript has been improved, and the reviewers are satisfied with most of the changes, but there is one remaining issue that needs to be addressed, as outlined below:Two reviewers with expertise in cryo-EM both have reservations about the updated cryo-EM data processing and remain concerned about the model fitting to the map. However, the reviewers think that the manuscript still provides new, valuable insight even without the cryo-EM data and that this data is not essential to support the main conclusions. Thus, the cryo-EM data should be removed from the manuscript, and the conclusions and models should be rewritten accordingly.

We have removed the cryo-EM data in the Results section, starting at line 417, and in the Methods section starting at lines 785 and 815 (details on sample preparation), and line 1045 (details on data processing). The removal of the cryo-EM Figure 5 (and supplementary) has set up a figure number shift from Figure 6 to 9, which are now called Figure 5 to 8. Furthermore, we have removed the previous Figure 7D, so now, the updated Figure 6 only contains A to C. We have also introduced a minor change to Figure 9 (now called Figure 8). These changes have their impact in the corresponding figure legends.

In addition, we have made some changes in the Discussion section (paragraph staring at line 564) to remove the final model that was hypothesized taking into account the cryo-EM data, and rewritten the conclusions accordingly.